# Monarch Butterflies in Western North America: A Holistic Review of Population Trends, Ecology, Stressors, Resilience and Adaptation

**DOI:** 10.3390/insects15010040

**Published:** 2024-01-07

**Authors:** David G. James

**Affiliations:** Department of Entomology, Washington State University, Irrigated Agriculture Research and Extension Center, Prosser, WA 99350, USA; david_james@wsu.edu

**Keywords:** population dynamics, stressors, neonicotinoids, climate change, habitat loss, resilience, adaptation, extinction of experience

## Abstract

**Simple Summary:**

The iconic monarch butterfly has undergone a dramatic decline in western North America since the 1990s, but in recent years has shown a capacity for resilience. This review identifies and investigates the likely drivers of this decline and resilience, and discusses their relative importance from a holistic viewpoint. Pesticides (particularly neonicotinoid insecticides), climate change, and habitat loss are likely to be the prime drivers of contemporary decline and instability in monarch population dynamics. Natural enemies (predators, parasites, and pathogens) are less likely to be a major contributor to contemporary population fluctuations, except on a local scale. Adaptation to changing environmental conditions will be an important component of the ongoing ability of western monarchs to show resilience. Human involvement with monarchs is a good and necessary thing for sustainable conservation, helping to prevent the ‘extinction of experience’, and the loss of human-nature contact, which has major adverse implications for nature conservation generally.

**Abstract:**

Monarch butterfly populations in western North America suffered a substantial decline, from millions of butterflies overwintering in California in the 1980s to less than 400,000 at the beginning of the 21st century. The introduction of neonicotinoid insecticides in the mid–1990s and their subsequent widespread use appears to be the most likely major factor behind this sudden decline. Habitat loss and unfavorable climates (high temperatures, aridity, and winter storms) have also played important and ongoing roles. These factors kept overwintering populations stable but below 300,000 during 2001–2017. Late winter storm mortality and consequent poor spring reproduction drove winter populations to less than 30,000 butterflies during 2018–2019. Record high temperatures in California during the fall of 2020 appeared to prematurely terminate monarch migration, resulting in the lowest overwintering population (1899) ever recorded. Many migrants formed winter-breeding populations in urban areas. Normal seasonal temperatures in the autumns of 2021 and 2022 enabled overwintering populations to return to around the 300,000 level, characteristic of the previous two decades. Natural enemies (predators, parasitoids, parasites, and pathogens) may be important regional or local drivers at times but they are a consistent and fundamental part of monarch ecology. Human interference (capture, rearing) likely has the least impact on monarch populations. The rearing of monarch caterpillars, particularly by children, is an important human link to nature that has positive ramifications for insect conservation beyond monarch butterflies and should be encouraged.

## 1. Introduction

The Monarch butterfly, *Danaus plexippus* (Linnaeus, 1758), in western North America appears to be undergoing a period of flux in terms of population size and ecology. From a historical high of an estimated three to ten million butterflies overwintering annually along the California coast in the 1980s, the population fell precipitously to about 200,000–400,000 overwintering butterflies at the beginning of the 21st century [1]. For 16 years (2001–2017), winter populations ranged from about 58,000–300,000 butterflies before another substantial decline to around 30,000 butterflies occurred in 2018. The smallest overwintering population of monarchs ever recorded in western North America occurred in 2020 (1899), before a remarkable rebound to 247,246 in 2021 and 335,479 in 2022 [https://www.westernmonarchcount.org/ accessed on 2 January 2024].

Considerable debate among scientists and non-scientists as to the driving forces behind these dramatic population dynamics has ensued, particularly over the past five years when extirpation of the western monarch was considered a possibility [1,2,3]. Our knowledge of the biology and ecology of monarchs in western North America is substantially less than our knowledge of the eastern North American population which has been the focus of extensive research for more than sixty years [4,5,6,7]. Thus, our baseline of what is ‘normal’ for the west is less certain, making it more difficult to assess the significance and importance of contemporary volatility in monarch population dynamics.

There is broad agreement that monarch populations in the west are affected by multiple stressors, but opinions differ greatly in assigning relative importance to individual constraints on population success. The aim of this narrative review is to provide a background to the population dynamics of monarch butterflies in western North America during the past forty years and to examine the likely influences and relative importance of biotic and abiotic stressors in determining population trends. My intention is to provide a holistic analysis that considers all potential stressors and assigns relative importance based on our knowledge of monarch biology and ecology. Most of the literature on monarch biology, ecology, and conservation with relevance to western North American populations, published during the past 50 years, was examined. A Google Scholar search conducted in June 2023 identified 2030 peer-reviewed articles containing the words “*Danaus plexippus*”, “Western North America”, and at least one of the following words: “population”, “ecology”, “stressors”, and “conservation”. Of these, 312 were research papers that were used in this review, along with many citations included within them.

## 2. Monarch Population Dynamics in Western North America (1980s–2022)

### 2.1. Monitoring

In common with the eastern North American population of monarchs, the size of aggregated overwintering populations has been used to provide an estimate of the population size of western North American monarchs [1,2]. However, unlike the eastern population, most of which overwinter in a single geographic area in Mexico [4], the western population overwinters at almost 400 widely separated sites along 1100 km of the California coastline [8,9]. During the 1980–1990s, estimates were made at small numbers (1–40) of overwintering sites as part of scientific studies [10,11] or limited citizen scientist efforts [1]. In 1997, a more organized count conducted by citizen scientists was established by Walt Sakai, Dennis Frey, Mia Monroe, and David Marriott [12]. They proposed to conduct a count over a three-week period every Thanksgiving at 100 or more overwintering sites. The Western Monarch Thanksgiving Count (WMTC) was adopted by the Xerces Society for Invertebrate Conservation in 2000 and since 2016 has included counts from 253–272 overwintering sites annually. Citizen scientists are trained annually to count monarchs roosting at overwintering sites, but the numbers are estimates at best because of the density of individuals in clusters. Nevertheless, these ‘counts’ are more accurate than the estimates of overwintering monarchs in Mexico derived from the number of hectares that they occupy [13]. Ultimately, the WMTC provides us with a proxy for the size of the western monarch population since at least 1997. Less confidence can be applied to pre-1997 estimates, but they still provide a useful ‘snapshot’ of monarch populations during the 1980–1990s.

### 2.2. The Big Decline Post-1997

Schultz et al. [1] estimated that the monarch overwintering population at the turn of the century had fallen by 97% of its average historic abundance in the 1980s from about 3–10 million to 200,000–400,000 butterflies. Thereafter, for 16 years until 2017, the overwintering population fluctuated within a range of 100,000–300,000 individuals (Figure 1). The sudden drop and consistent lower overwintering numbers after 1997 suggest a major detrimental change within the ecology of western monarchs occurred at that time. A 67% drop in the eastern North American overwintering population in Mexico occurred one year before the 55% drop in the western population, suggesting some commonality of causative factors. Many monarch scientists believe that the rapid increase in the use of genetically-modified herbicide-resistant crops, primarily soybean and corn, over vast areas of the eastern US from 1996 onwards was a prime driver of the decline in overwintering populations in Mexico [14]. For the first time, weeds could be sprayed without affecting crops, resulting in large cropland areas, particularly in the mid-west, devoid of weeds, including milkweed [15]. While herbicide-resistant crops were also used in the western US, croplands occupy a much smaller area west of the Rocky Mountains [16].

### 2.3. Neonicotinoids as the Potential Primary Driver of the Western Monarch Decline Post-1997

The major stressors on monarch populations in North America are widely considered to be habitat loss, climate change, and increased use of pesticides [5]. Habitat loss is considered to be an important driver of monarch decline in North America [17], but its impact is likely to be relatively progressive over time and not sudden. Similarly, any adverse impact of climate change on monarch ecology would likely be progressive and not sudden. While the growth in the use of synthetic pesticides has also been progressive since the 1950s, a new class of pesticides, the neonicotinoids, emerged in the early 1990s to rapidly become the most widely used insecticides in the world [18]. Neonicotinoid use in North America increased dramatically from 1994–2011 [19], coinciding with a 55–67% decline in the size of monarch overwintering populations. Today, neonicotinoids still maintain the largest global share of the insecticide market (24%) [20].

While any association between monarch population decline and the rise of neonicotinoid use in the late 1990s remains correlative, there is now much field and laboratory evidence for lethal and harmful sub-lethal effects from this class of insecticides on beneficial insects, including butterflies [21,22,23]. Correlative associations have also been reported for population declines of butterfly faunas in Great Britain and California and the increased use of neonicotinoid insecticides [24,25]. While most studies on the sub-lethal impacts of neonicotinoids have focused on honey bees [26,27], an increasing number have looked at butterflies [28]. Sublethal impacts of field-realistic levels of the neonicotinoids, imidacloprid, and clothianidin on monarch larvae were reported by Pecenka and Lundgren [29] and Krischik et al. [30]. Clothianidin at levels of 0.5–5.0 ppb reduced the weight and body length of first instar larvae. It also increased the period spent as first instar larvae and resulted in 50% mortality [29]. Krischick et al. [30] showed imidacloprid at 15 ppb significantly reduced survival of early instar monarch larvae. Arrested pupal ecdysis occurred in 60–82% of monarch larvae exposed to neonicotinoids as late instar larvae [31,32].

Neonicotinoids are highly systemic compounds readily transported through the vascular system of plants, poisoning herbivores whether they feed on stems, leaves, flowers, or seeds [33]. A likely important route of neonicotinoid exposure to pollinators like butterflies is dissemination through nectar and pollen [34]. In addition to crop flowers being contaminated with neonicotinoids, the mobility and longevity of these compounds in soil and water [35,36] means that residues can also be found in non-target plants at distances from crops [37]. Surveys of streams in agricultural and urban areas in the US have found neonicotinoid residues widespread in surface waters [38,39,40], and they have also been found in snow and spring meltwater in Canadian prairie wetlands [41]. Neonicotinoid insecticide residue levels in crop nectar and pollen differ considerably according to the amount applied to crops and landscapes as well as the method of application. Seed treatments result in relatively low levels of nectar and pollen, usually less than 10 ppb [34,37]. Residues in pollen and nectar from crops treated with foliar applications range from 10–100 ppb [30]. The greatest neonicotinoid residues (1000–4500 ppb) are found in nectar and pollen from landscape trees and plants treated with soil drenches [30]. Another less-researched, but likely source of high levels of neonicotinoid contamination of pollen and nectar, are home garden plants [42], which may receive recommended application rates > 40 times greater than those used in agricultural systems. Garden plants propagated in nurseries receive even higher rates of application and may contain neonicotinoid residues in nectar and pollen of up to 45,000 ppb [30]. Recent research indicates that plant species vary in the efficiency of uptake of neonicotinoids. Milkweeds (two species) had a low uptake (<0.5%) compared to Red Clover (50%) [43].

A brief laboratory study assessed the impact of the neonicotinoid, imidacloprid, provided as adult nourishment on adult monarch longevity [44]. Imidacloprid at 23.5 ppb, a field-realistic rate reported from wild nectar and pollen, was fed ad libitum to newly-eclosed monarchs in a sugar-based diet for 22 days. Treated monarchs showed reduced longevity, with 78.8% mortality by day 22, compared to 20% in untreated monarchs. In a similar study, imidacloprid/syrup, force-fed at 15 and 30 ppb, had no effect on the survival of monarchs [30]. In addition, free-ranging monarchs in small mesh cages allowed to feed on flowers containing 6030 ppb or 10,400 ppb also showed no difference in survival and fecundity from non-exposed butterflies [30]. However, the authors noted that monarchs may not have been able to forage adequately in the small cages. Uncontaminated 30% honey water sponges were also supplied as food, and monarchs may have fed less on or avoided the neonicotinoid-contaminated flowers. More recently, Prouty et al. [45] concluded that adult monarchs show high tolerance to field-realistic levels of neonicotinoid insecticides. However, these authors held monarchs for only 10 days during exposure to imidacloprid, which is insufficient to show an impact on medium- to long-term survival, judging from the results of James [44]. Similarly, Krishnan et al. [31] held neonicotinoid-treated monarchs for just four days to assess mortality. Mortality in monarchs exposed to field-realistic levels of imidacloprid occurred during days 12–22 [44]. If laboratory insecticide bioassays are to reflect real-world scenarios, then materials used in the field should be tested. Thus, commercially formulated insecticide products [44] rather than active ingredients [30,31,45] should be used. Commercial formulations of insecticides have invariably proven to be more toxic than active ingredients alone [46,47,48]. More work is needed on the impact of nectar-borne neonicotinoids (commercial formulations) on the medium- to long-term survival of adult monarchs. The possibility of ingested neonicotinoids during adult life impacting the survival and development of progeny should also be investigated.

The extent to which neonicotinoids currently contaminate the wider landscape is an important question. The ability of neonicotinoids to move away from sites of application through water, air, and soil [41,49], combined with residual persistence [50], provides the potential for widespread landscape contamination. The contamination of waterways and urban water supplies by neonicotinoids is increasingly being documented [40,51,52] and is concerning not only for pollinators but also for human health [53,54,55]. The ubiquity of neonicotinoids as ingredients in home garden pest control treatments, along with the higher rates used for these applications, has the potential to make urban areas highly contaminated with neonicotinoids. Gardens not directly exposed to neonicotinoid sprays may still be contaminated by run-off from other gardens that use these compounds. Urban areas may be reservoirs of high-level neonicotinoid contamination [56].

Shortened adult longevity has serious consequences for monarch population development, migration, and overwintering. Although egg production appeared to be unaffected by adult exposure to imidacloprid [44], we do not know whether the viability of eggs and resultant larvae is unaffected. Mating behavior could also be affected, as shown in a parasitic wasp species [57]. Monarch migration is powered by feeding on a wide range of nectar in natural, agricultural, and urban landscapes, at least some of which are likely to be contaminated with neonicotinoids. Could the migratory ability of monarchs be affected by these residues? Neonicotinoid-mediated impairment of foraging behavior has been reported for bumblebees [58]. The flight behavior of locusts (*Locusta migratoria* L.) is impaired by imidacloprid [59], and the migration and metabolism of some birds also appear to be impaired by this chemical [60,61,62].

Correlative declines in butterfly [24,25] and bird faunas [63,64] have been associated with the introduction and widespread adoption of neonicotinoids. The relatively abrupt decline in western Monarch populations between 1997 and 2001 may have been another example of this.

### 2.4. Sudden Decline 2018–2019

After a 16-year period (2001–2017) of relatively stable overwintering populations fluctuating between 100,000–300,000 butterflies, there was an 86% decline between 2017 (192,624) and 2018 (27,721). A similarly low estimate was made in 2019 (29,436). This sudden drop was considered to mirror a ‘textbook extinction vortex’ [2]. These authors restated a 2016 proposition that 30,000 butterflies represented the quasi-extinction threshold for western monarchs [1,2]. 

What was the cause of this sudden and rapid decline? Mid-to-late winter (January–March) in 2017, 2018, and 2019 was characterized in coastal California by significant winter storms with substantial rainfall and high winds (e.g., https://www.climate.gov/news-features/event-tracker/soaking-rains-and-massive-snows-pile-california-january-2017, accessed on 2 January 2024) which caused serious damage to some overwintering sites. Monarch populations in late winter are most vulnerable in terms of physical wear and tear to wings and lipid depletion, and it is possible that much mortality and/or early dispersion occurred during these winters, as has been recorded for overwintering monarch populations in Mexico [65]. Substantially earlier dispersal of butterflies was recorded at Lighthouse Field in Santa Cruz, with an 80% decline in January 2017 (J. Dayton, pers. comm). Dispersal at this early time may limit the reproductive potential of females. The first storm event in January–February 2017 was followed by a 35.5% overwintering population decline from 298,464 in 2016 to 192,624 in 2017. The storms during January–March 2018, were followed by an 86% decline to less than 30,000 overwintering butterflies in 2018/19 (Table 1). Although no data are available on the size of reproductive populations in the summers following these severe winter storm events, it is possible that the winter storms caused the decline. An overwintering population of ~200,000 may not be large enough to provide a good buffer against winter storms which may be increasing in frequency and intensity in California [65].

### 2.5. 2020: The Western Monarch Nadir

2020 was the year that the ‘extinction vortex’ seemed to have become a reality for the western monarch butterfly. With just 1899 monarchs counted at 249 overwintering sites, the population declined by 93.6% in a single year (Table 1). Populations at overwintering sites ranged from 1–550 and only 100 sites actually hosted butterflies [https://www.westernmonarchcount.org/ accessed on 2 January 2024]. The western monarch population, at less than 0.5% of its 1980s peak, was considered on the ‘brink of collapse’ with the possibility that they could be lost from the interior west [3]. 

In October–November 2020, the Washington State University monarch tagging program recorded some unusual data. For the first time, no tagged butterflies were recovered from overwintering sites. During 2017–2019, 67% of recovered monarchs tagged in the Pacific Northwest were found at overwintering sites [66]. Of about 1300 monarchs tagged in late summer-fall 2020 (mostly in southern Oregon), all recoveries (10) were made in northern CA or the San Francisco Bay area, in association with host plant milkweeds. These butterflies, after traveling 300–500 km, appeared to have become reproductive. A precedent for this was seen in 2017 and 2019, when two tagged females from Oregon were observed laying eggs in Santa Barbara [67] and San Francisco after migrating 877 and 537 km, respectively [66]. James [67] suggested that migrants becoming reproductive might contribute to the population decline at overwintering sites.

In December 2020, it was also apparent that citizen scientists were finding large numbers of monarch eggs and larvae in the San Francisco Bay area. An analysis of the number of sightings of monarch larvae and pupae in the San Francisco Bay area in January 2021 showed more than three times as many larvae and pupae reported than in the same month over the previous six years (Figure 2) [68]. 

A study of monarch winter breeding in the San Francisco Bay area during January–April 2021 showed that adults were common during February-March with numbers ranging from 0.23–1.54/min during ~30 min weekly surveys [69]. Eggs and larvae were abundant during the same period. The convergence of data on the absence of butterflies at overwintering sites and the presence of novel breeding populations in the San Francisco Bay area (and anecdotally in other near-coastal urban areas of California, including Los Angeles), indicated a possible shift in overwintering strategy by western monarchs. 

Monarch populations in south-eastern Australia in the early 1960s overwintered as non-reproductive populations comparable to California (up to 40,000 butterflies/site) in the Sydney Basin, New South Wales [70]. Research on these populations during 1978–1984 showed the existence of synchronous reproductive and non-reproductive overwintering populations [71,72,73]. Coincident with this apparent shift in overwintering behavior was a fall in the size of overwintering colonies from 40,000 to a maximum of 3500 butterflies per site [72]. This 90%+ decline in overwintering monarch populations was thought to have resulted from the loss of milkweed habitat, but it is possible that the shift towards winter-breeding was at least partly responsible. 

If monarch butterflies migrating through the San Francisco Bay area during September 2020 became reproductive and stopped migrating to overwintering sites, as the tag recovery and iNaturalist data indicate, why did this happen? Migrating monarchs in eastern North America are in reproductive diapause [74], which means diapause usually cannot be broken until a period of refraction (when there is no response to temperatures normally stimulatory to reproductive tract development) has passed, normally in mid-winter [75]. While this may apply to most of the population and monarchs arrive at the Mexican overwintering sites with undeveloped reproductive tracts [4], some do break diapause in the southern US and drop out of the migration [76,77]. Australian monarch butterflies possess a flexible reproductive dormancy, oligopause, that has no refractory period [78]. Exposure of reproductively dormant Australian monarchs to temperatures optimal for reproduction at any time results in reproductive tract development. The nature of reproductive dormancy has not been explored in western monarchs but might be expected to be diapause, given that the North American population is genetically homogenous [79]. However, some environmentally induced biological differences have been detected in the western population [79], and reproductive dormancy in western monarchs may be more labile than in the eastern population. 

It does seem likely that reproductive dormancy and migratory behavior were terminated prematurely in many western monarchs arriving in California in the autumn of 2020. Temperature is the main driver of reproductive development in adult monarchs [78,80], and the San Francisco Bay area experienced record-breaking temperatures during September–October 2020. The mean daily maximum temperature for San Francisco during September–October 2020 was 24.7 °C, 2.5 °C above the historical mean, and migrants were exposed to temperatures up to 39 °C [68]. If some migrants became reproductive in autumn 2020, forming a substantial winter-breeding population in the San Francisco Bay area, then the official WMTC count of 1899 underestimated the size of the western monarch population. Crone and Schultz [3] estimated a summer breeding population of about 12,000 monarchs in the Bay Area, and it seems likely there were at least that many, if not more, in the Bay Area during winter 2020/21. 

### 2.6. 2021: Monarch Resurrection

In February 2021, the result of a New Year count of the overwintering population was released, showing that the already-small population in late December/early January had fallen further to 1069 butterflies [https://www.westernmonarchcount.org/ accessed on 2 January 2024]. If we assume a 50:50 sex ratio, then there were just 535 females remaining to begin the development of the 2021 summer population. However, as indicated above, a likely substantial population of monarchs also existed as breeding populations in near-coastal urban California, from San Francisco to Los Angeles.

The extent and success of the first generation of eggs and larvae produced in March–April by surviving overwintered females and females in winter-breeding populations in 2021 is unknown. This generation develops within California, and it is notable that the intensity of breeding at South Bay winter-breeding sites increased during March [69]. Adults from this first spring generation migrate during May and June into far northern California, Oregon, and Washington. The number of monarchs seen in Oregon, Idaho, and Washington during migration in April–June 2021 reported to websites like iNaturalist, was the same (25) as during the same period in 2020 [James, unpubl. obs]. This was surprising given the difference in the size of overwintering populations (WMTC January counts) that spring migrants were derived from in 2019/2020 (11,971) and 2020/2021 (1069). This is likely further evidence that the ‘true’ size of overwintering populations in 2019/2020 and 2020/2021 may have been comparable if winter-breeding populations were included. During July and August 2021, 43 monarchs were reported from the Pacific Northwest, almost double the number reported during the same period in 2020 (22). 

The summer 2021 population of monarchs in the west appeared robust enough to improve populations at the traditional overwintering sites in winter 2021/2022 as long as migration was strong and temperatures during September–October in California were seasonal. The mean daily maximum temperature during September–October 2021 in San Francisco was 21.3 °C, below the long-term mean of 22.2 °C for this period (Figure 3). 

Consequently, monarchs migrated through the Bay Area and other parts of California, likely with minimal loss of individuals to reproductive populations. The number of larvae and pupae reported in the San Francisco area during January 2022 to iNaturalist was substantially lower than in 2021 (Figure 2), indicating a lower incidence of winter breeding. 

The overwintering monarch population in California increased from 1899 in 2020 to 247,246 in 2021 [https://www.westernmonarchcount.org/ accessed on 2 January 2024], a 130-fold increase (Table 1). Simple mathematics shows that a population of 535 females leaving overwintering sites in late winter 2021 could not result in a population increase of this size. Even with overly-optimistic assessments of development and survival, a population of between 10,000 and 20,000 post-overwintering females would be needed to produce a population of 300,000+ in one season [Chip Taylor, pers. comm.]. This is further evidence that there were other females (e.g., those in winter-breeding populations) in early 2021 to help produce the scale of increase that occurred during the summer. It seems likely that winter-breeding populations were part of this, but it is also possible that some migrants reacted to the above-average warm-hot conditions of September–October 2020 by forming small, inconspicuous, non-reproductive overwintering populations at undiscovered sites, perhaps in the higher-elevation coastal range of California. There is precedent for this, with monarchs sometimes forming temporary roosts in hot autumn conditions in Texas [Chip Taylor, pers. comm.] and in Australia [81]. Another possibility for increasing the western population during a single season could come from spring migrants leaving overwintering populations in Mexico and arriving and laying eggs in Arizona [82]. Most likely, winter-breeding populations, undiscovered overwintering butterflies, and a spring incursion from Mexico all contributed in some way to the recovery of western monarchs in 2021. 

Pacific Northwest summer populations in 2022 as judged from iNaturalist reports and personal communications appeared to be almost 10-times greater than in 2021 [James, unpubl. obs.] The mean daily maximum temperature for San Francisco during September–October 2022 (22.1 °C) was close to the long-term average (22.2 °C) (Figure 3), and the overwintering population grew by about 35% to 335,479 [https://www.westernmonarchcount.org/ accessed on 2 January 2024] (Table 1). 

### 2.7. Western Monarchs: The Future

It is likely that there will be wide swings in abundance as the monarch responds to a changing environment, primarily the climate. We should also expect adaptation and resilience to changing conditions. From their arrival in Australia in 1872 [83,84], monarchs adapted to a new environment by substantially changing their physiology and behavior [85]. Migratory butterflies like the monarch have great genetic diversity and a better ability to adapt and produce larger populations [86].

California’s climate has demonstrably warmed during recent decades [87,88] and may have reached a tipping point in terms of facilitating ‘normal’ migratory and overwintering behavior for monarch butterflies. Autumn temperatures in California may now determine whether the majority of migrants overwinter at traditional sites in reproductive dormancy or develop winter-breeding populations in near-coastal urban areas. Warmer conditions may enable monarchs to breed during the winter in coastal areas north of San Francisco. Since the 1960s, we have seen a northward spread of monarch winter-breeding from San Diego to San Francisco [69,89]. A warming climate is also predicted to result in more frequent and intense winter storms in California [90], which will increase monarch mortality at traditional overwintering sites (but not necessarily at winter-breeding sites). Traditional overwintering populations in the future will be the product of interaction between higher fall temperatures, an increased frequency of winter storms, and the ability of monarchs to cope with and adapt to these environmental changes.

Future Thanksgiving counts at traditional overwintering sites may not always provide a realistic assessment of western monarch population size and trends, depending on the proportion that develops winter-breeding populations. There may also be a shift in occupied overwintering sites towards the north or to higher elevations as winter temperatures rise [9]. The future of Monarch populations in western North America will depend on how successful the butterfly is in responding and adapting to a changing environment, primarily a warming climate. 

## 3. A Holistic View of Western Monarch Population Stressors

Monarch butterflies in western North America are exposed to multiple stressors that cause varying degrees of mortality in the population. The relative importance of these stressors is a subject of much debate. However, there is good consensus among researchers that pesticide exposure, changing climate, and habitat loss are the ‘big three’ stressors for monarchs throughout their North American range [5]. 

### 3.1. Pesticides

The likelihood that the increasing use of pesticides in agriculture and urban areas is a significant driver of long-term monarch population decline was discussed above, with particular reference to neonicotinoid insecticides (2.3). There are other classes of commonly used insecticides that also have deleterious impacts on non-target organisms including butterflies [28]. Synthetic pyrethroid insecticides are the second most frequently used insecticide class (15% market share) after neonicotinoids and pose substantial lethal and sub-lethal risks to monarchs [91,92]. Organophosphorus insecticides are highly toxic to most insects including butterflies, and are still used in some agricultural situations [93,94]. Even some biopesticides, touted as being safe to many non-target organisms, can be lethal for butterflies including monarchs. A major biopesticide used commonly in agriculture, forestry, and home gardens is *Bacillus thuringiensis* (*bt*), a soil-dwelling bacterium [95]. It also occurs in the gut of caterpillars and is an important natural pathogen regulating butterfly and moth populations [96]. Spray formulations of *bt* applied to crops and forests have the potential to cause substantial mortality to non-target butterfly and moth larvae including monarchs [97]. *Bacillus thuringiensis* accounts for 80% of all biopesticides used in the US on crops with millions of acres of farmland, forest, and urban areas treated annually.

Many new classes of insecticides are being developed [98] and we have little information on the impacts of these on non-target insects including butterflies [99]. For example, the only research published to date on diamide insecticides which were introduced in 2008 and now have a global market share of 12% [20], showed chlorantraniliprole to be highly toxic to all life stages of monarchs [31]. There are six other insecticides in the same class for which we have no information about their impact on monarchs or any other butterfly. The same is true for at least six other classes of novel insecticides (sulfoxamines, butenolides, pyropenes, mesoionics, isoxazolines, and ethylsulfones). 

Fungicides and herbicides are rarely lethal to butterflies but have been shown to have deleterious sub-lethal impacts. Some fungicides appear to reduce wing length in monarchs [100] and some herbicides have been implicated as a possible cause of direct decline in some butterflies [101]. The indirect effect of herbicides limiting monarch abundance by the removal of milkweeds from the landscape is considered to be an important driver of monarch decline in eastern North America [15]. While milkweeds in the eastern US commonly grow in large acreage croplands, this is not the case in the arid west. Milkweeds in the west are patchily distributed, occurring mostly in riparian, roadside, and urban habitats, and are less likely to be affected by herbicide use in agriculture [102,103]. 

The extent of pesticide contamination of monarch habitats in an agricultural landscape in the west was indicated in a recent study [104]. Samples of milkweed leaves from 19 sites in the Central Valley of California showed contamination by 64 pesticides (25 insecticides, 27 fungicides, and 11 herbicides). All samples had at least one pesticide present and on average nine compounds were found per plant. Chlorantraniliprole, known to be toxic to monarchs [31], was identified in 91% of samples. The impact on monarchs of most of the 64 pesticides identified in this study, is unknown. Also unknown is the additive or synergistic effect of these compounds on monarchs when present in a mixture. Comparable studies on pesticide contamination of milkweeds in non-agricultural or urban areas of the west are not available. Pollinators are exposed to and assimilate pesticides other than neonicotinoids [105]. Native bees and butterflies collected from margins of agricultural fields contained nine pesticides including insecticides, herbicides, and fungicides. Sampled monarchs contained bifenthrin (synthetic pyrethroid insecticide), imidacloprid (neonicotinoid insecticide), and tebuconazole (fungicide). Even the long-banned DDT and a metabolite were found in monarchs [105]. Aside from imidacloprid, we have no information on the possible sub-lethal effects of these other pesticides on monarchs. Urban areas are likely to be important breeding areas for monarchs in the west [69]. These are also areas where much pesticide use occurs in home gardens, parks, sports fields, golf courses, and other public and private spaces. The rates of use of many pesticides like neonicotinoids are greater for urban applications compared to agricultural use [30]. Growing milkweeds in gardens and yards is one action that the public can do to help monarch populations. However, sourcing milkweed plants from nurseries can also expose caterpillars to pesticides. Halsch et al. [106] detected 61 pesticides with an average of 12.2 compounds per plant, from 33 retail plant nurseries from across the US. This study, while alarming, only looked at pesticide residues in milkweed leaves. Presumably other plants sold by nurseries including butterfly nectar plants also contain pesticide residues and in the case of systemic and persistent materials like neonicotinoids, may be present in nectar for a long time after purchase.

### 3.2. Climate

Climate is a large-scale geographic stressor on monarch populations. However, climate change is not occurring at the same rate everywhere. Increasing temperatures particularly in winter and spring are a feature of climate change in the western US [107,108]. This has resulted in intense and widespread droughts [87,88] as well as increased incidence and severity of winter storms and flooding [90,109]. Droughts, winter storms, and flooding impact monarch populations, mostly in a deleterious way. For example, drought was negatively correlated with monarch abundance in the west [102]. 

When daily maximum temperatures exceed 35–38 °C for sustained periods, developing eggs, larvae, and pupae as well as adult monarchs suffer. Immature stages suffer increased mortality from temperatures of 36 °C and above depending on the length of exposure and stage [110]. A constant temperature of 42 °C for 12 out of 24 h for two days resulted in 80–90% mortality of early and late instar larvae [111]. A constant temperature of 38 °C for 12 out of every 24 h for six days resulted in 30% mortality of third instar larvae [111]. Prolonged heatwaves have become more frequent and intense in the western US [112]. James [113] documented an apparent heatwave-induced reduction in a summer-breeding population of monarchs in July 2015 in central Washington. Fifteen consecutive days with daily maximum temperatures above 38 °C, led to an apparent 75% reduction of monarchs, likely due to increased mortality of developing eggs and larvae. Increased frequency and intensity of heatwaves in the future will lead to a reduction in optimal landscapes and habitats for monarch breeding in the western US. An increase in wildfire activity across the west is another consequence of climate warming, heatwaves, and drought [107]. Wildfires directly reduce monarch habitat and populations temporarily but may improve milkweed habitats in the long term [114]. The increase in wildfire smoke may lead to reduced air quality [115] which may also be detrimental to monarch survival and migration. A limited analysis [66] showed no apparent impact of smoky conditions on the migration of a small sample of western monarchs and their longevity. A warming climate also appears to be disrupting the phenology of western monarch migration and breeding (see Section 2.5). Winter-breeding areas for monarchs in the west have within 50 years spread from being limited to the San Diego area [89] to now occurring in the San Franciso Bay area [69]. A detailed understanding of how climate warming affects the phenology of migration and breeding in western monarchs will be challenging [116]. 

An increased frequency of strong winter storms in California appears to be associated with a warming climate [90,117]. In 2018, 2019, and 2023, strong winter storms occurred mid-late winter (January–March). Overwintering monarch populations are vulnerable at this time because of energy reserve depletion [11] and because they are about 40% smaller than in November [https://www.westernmonarchcount.org/ accessed on 2 January 2024]. Winter storms in California are invariably associated with strong winds, and clustering monarchs at overwintering sites can be torn from the tree branches and foliage that they roost on [118]. Survival is dependent on the length of time it takes them to crawl up or fly from the ground. Overwintering populations at some sites appear to be especially vulnerable particularly if the site is in a gully, close to the ocean, and oriented east-west. Other sites, further inland, on elevated ground, within woodland, or with natural windbreaks, may offer better protection [119]. Although a changing climate is clearly having an impact on western monarch populations, at least two studies concluded that it does not appear to be the major driver in population declines, with trends in abundance more strongly associated with land use [120,121].

### 3.3. Habitat

In contrast to eastern North America, little is known about the character, distribution, importance, and status of monarch habitat in the west. Stevens and Frey [102] initially clarified the summer-breeding habitat of monarchs in the west and this was later improved by Dilts et al. [122]. Strong support for changes in land use as a cause of monarch population decline has been reported [121], but these authors also found that unambiguous separation from climate and pesticide use was difficult. 

#### 3.3.1. Spring Habitat

Western monarch populations appear to be at their most vulnerable in late winter-early spring [120]. A problem dispersing females face is finding milkweed for egg laying. This necessitates migration of at least 10–20 km and sometimes up to 200 km to find suitable patches of newly sprouting milkweed. Generally, milkweed does not occur close to overwintering sites, and cultivation of milkweed within five miles of overwintering sites is discouraged by conservation organizations [https://xerces.org/sites/default/files/publications/19-004.pdf accessed on 2 January 2024]. The poor condition of overwintered butterflies and the lack of and difficulty in finding milkweed means that many females from overwintered populations may not achieve optimal egg production. Early season native milkweeds (e.g., *Asclepias californica* Greene, *A. cordifolia* (Benth.) Jeps, or *A. eriocarpa* Benth.) may be uncommon and when overwintering colonies break up early because of storms or warm temperatures, these milkweeds, mostly in mid-high elevation areas, may not be available to egg-laying females. Recognition of this early-season habitat problem for monarchs has led to a focus by conservation organizations on improving it, by planting early-season native milkweeds and enhancing available habitat [2]. The increased cultivation of ornamental, non-native milkweeds in near-coastal urban areas of California over the past decade [69] may help to alleviate the early spring lack of milkweeds, but this is not condoned or supported by some monarch scientists and conservation organizations (see Section 3.5) [2,123]. Getting the population off to a good start in California in early spring is one of the most critical components of monarch ecology in western North America. The first generation of new adults, the progeny of the overwintered population, migrate north and east for hundreds of kilometers during May and June, reaching into Oregon, Nevada, Washington, Idaho, and British Columbia. These migrants face challenges including adverse weather, exposure to predators, and road traffic mortality. The latter may be significant because spring migrants tend to fly low and are vulnerable to collisions with vehicles. In some years, evidence of this is notable along spring migration routes (James unpubl. obs.).

#### 3.3.2. Summer Habitat

The summer breeding habitat of monarchs in western North America is different from eastern North America. Aridity and mountains restrict and limit the extent of land (primarily riparian and riverine landscapes and flood plains) that provides suitable monarch habitats. In contrast, most of the landscapes in the eastern US have greater rainfall which enables monarch habitats to exist in all landscapes except densely forested or wooded areas. Consequently, milkweed is scattered across the landscape to a far greater degree than in the west. Milkweed and monarch habitat in the west tends to be confined to riparian, agricultural, and urban areas where there is sufficient moisture. Although studies have been published on the actual and potential distribution of suitable habitats for monarchs in the west [102,122,124], few published studies focus on monarch ecology in western US habitats [113,124,125]. The greatest continuous expanse of suitable habitat for western monarchs occurs in California but there are also large areas elsewhere in the west that contain high-quality monarch and milkweed habitat [122]. In the Pacific Northwest, large areas of suitable habitat occur in western Oregon, central-southern Washington and along the Snake River plain in Idaho [122,124,126]. There are about 40 species of milkweed in western North America, but only a few occur widely and are abundant. The two commonest and most widely distributed species in the west are Showy milkweed (*A. speciosa* Torr.) and Narrow-leaved milkweed (*A. fascicularis* Decne.), and these likely support most western monarch populations. Although most summer-breeding monarch populations in the west occur in the Pacific Northwest, Nevada, Utah, New Mexico, and Arizona, some populations remain within California [120].

A site-specific study on summer-breeding populations of monarchs over three years in central Washington [113] revealed some characteristics that may be typical of breeding populations in the arid west. This riparian site contained thousands of milkweed (*A. speciosa*) plants in a 2.4 km^2^ area that annually hosted a large (300–400 individuals), resident breeding population during June–August. This differs from the eastern US where milkweed occurs in small, separated patches [15,127,128] which generally do not support resident populations. Instead, eastern US summer breeding monarch populations are usually low-density, and evenly distributed over the landscape. Discrete, high-density milkweed stands within matrices of grasslands, shrub-steppe, wetlands, and woodlands, also occur at other sites in Washington and Idaho [124], and likely support high-density monarch populations. 

Although no information exists on changing milkweed population densities in the west, it is likely that declines have been modest in the Pacific Northwest. Some milkweed is likely lost to agriculture and herbicide spraying of roadsides and irrigation canal channels. Herbicide application, mowing, and invasive plant species were identified as primary threats to milkweed habitat in Idaho and Washington in 2016–2017 [124]. Milkweed declines as seen in agricultural areas of eastern North America [129,130] may have also occurred in agricultural areas of California, but data are lacking. Agriculture, urban development, and roadway maintenance are likely to be the principal reasons for milkweed loss in California. In particular, the clearance of vegetation from roadsides for safety and fire prevention reasons is especially aggressive in California and parts of Oregon, with routine spring applications of herbicide. In contrast, the Washington State Department of Transportation actively maintains and manages extensive roadside milkweed populations along major roadways in eastern Washington.

Habitat stressors other than loss of milkweeds may exert pressure on western monarch populations. For example, shade and late summer nectar resources are important resources for breeding populations in eastern Washington [113]. Monarch summer breeding habitats in Washington, Idaho, and eastern Oregon frequently experience high temperatures (>35 °C) and shade is a necessary feature of these habitats [113,124]. Most oviposition by Washington monarchs in July–August occurs on shaded milkweeds under trees and bushes [113], [James, unpubl. obs.]. Most daytime roosting by summer monarchs in eastern Washington and Idaho in response to high temperatures occurs in Russian Olive (*Elaeagnus augustifolia* L.) trees [113,124]. This is likely due to the predominance of this invasive species in the riparian habitats occupied by monarchs. Native trees and bushes are used if present [113]. Breeding habitats in arid areas of the Pacific Northwest have fewer flowering native plants as the summer progresses. Milkweed is the primary nectar source in early-mid-summer [113,124], but when it is senesced in late July, other options may be limited. At the central Washington site studied by James [113], the only flowers available for monarchs during August-September were those of the invasive Purple Loosestrife (*Lythrum salicaria* L.). When heat and drought prevented *L. salicaria* from blooming, monarchs abandoned the site [113]. Native plants like Goldenrods (*Solidago* spp.) and Common Sunflower (*Helianthus annuus* L.) do bloom during late summer in these areas and are used by monarchs [124]. Introduction or restoration of these native plants to breeding sites that currently lack them should be a conservation priority for monarchs in the Pacific Northwest.

Agriculture occupies a large percentage of optimal monarch habitat zones in the west, principally, the Central Valley of California, the Willamette Valley of western Oregon, the Columbia Basin of central and southern Washington, and the Snake River Valley in Idaho. Consequently, the incorporation of milkweed and monarch habitats within agriculture is desirable. Agricultural production in the US is making progress in reducing chemical inputs and increasing the use of biological and cultural strategies for pest control [131]. Vineyards in the Pacific Northwest are at the forefront of this change using conservation biological control, native habitat restoration, chemical ecology, and minimal use of ‘soft’ pesticides as the basis of pest management programs [132,133,134]. In some cases, vineyards have become functional habitats for butterfly populations [135]. The two dominant milkweed species in the west (*A. speciosa, A. fascicularis*) attract substantial numbers of predatory and parasitic insects in central Washington [136] and are recommended for cultivation in or near vineyards to improve natural control of insect and mite pests while also serving as increased habitat opportunities for monarchs. This ‘Beauty with Benefits’ approach is utilized by some grapegrowers in southern Oregon and southern Washington and is expected to expand in the future. A recent survey of eastern US farmers indicated that at least 50% would be prepared to voluntarily incorporate milkweed plantings in their non-cropped farmland [137]. 

Roadside habitat for monarchs is considered important in eastern North America [138,139,140] and is likely to be important in the west. Milkweeds frequently occur as roadside plants but are often subjected to herbicides and/or mowing. Increased awareness of the value of roadsides for pollinator populations in general [141] has led to management that minimizes the impact on monarchs and other pollinators. However, improvements are still needed because herbicide treatment of roadside milkweed patches still occurs [124].

The increasing incidence of wildfire in the west [107] may temporarily remove habitat from use by monarchs but evidence suggests that in the long-term, fire may benefit milkweed and monarch populations. Monarchs were more abundant on Minnesota prairie with a history of prescribed burns than on unburned prairie [114]. Early summer wildfires may also create patches of late-emergent milkweed, providing nectar and host plant resources for late-summer butterflies [142]. Similarly, appropriately timed mowing of milkweed can stimulate re-sprouting, providing resources for late-season monarchs [143,144,145]. In addition to creating milkweed stems with young leaves that are attractive to ovipositing monarchs, there is evidence that disturbances like fire or mowing may reduce predator populations [146].

#### 3.3.3. Autumn Habitat

Autumn is also a critical time in the seasonality of western monarchs. The fall migration, with movement across the landscape for up to 1392 km [66], is a dangerous time, with threats ranging from attack by predators, mortality from roadway collisions, adverse weather, and lack of food and/or moisture. At an average rate of travel of 20.7 km/day [66], monarchs may be exposed to these threats for 5-to-10 weeks, as they make their way to overwintering sites in California. Nectar sugars converted to lipids fuel the migration [147], so the most important resource that migrating monarchs need is a sustained supply of nectar along the migration route. Although the routes that migrating monarchs take in the west are unknown, it is likely that migrants traverse arid and barren landscapes. However, yellow-flowering shrubs known collectively as ‘rabbitbrush’ primarily in the genera *Chrysothamnus* and *Ericameria* are common and widespread in the arid west, flowering during the monarch migration. There is no evidence that the quantity and quality of this likely major nectar source for migrating monarchs in the west has declined in recent decades. These flowers are frequently visited by monarchs and their ubiquity makes it likely that they play an important role in sustaining western monarch migration. Migrants have been observed flying down river valleys and through associated urban areas [148], and these will find additional flowering plants for nectar. Recent research on the nature of stored lipids in overwintering monarchs in Mexico suggests that most energy reserves are accumulated near the overwintering area [149]. If the same thing occurs in western monarchs, then most of the stored fat reserves needed for winter survival will be obtained from flowers in California, as they get close to the coastal overwintering sites. Increasing fall-flowering nectar plant abundance within 100 km of overwintering sites could be an effective conservation strategy. Increased mortality during migration has been suggested as a reason for the decline in the eastern North American monarch population [150]. However, a recent study using tagging data found no increase in migration mortality during 1998–2015 [151]. It is unknown if this is also the case for western monarchs.

#### 3.3.4. Winter Habitat

Overwintering of the western monarch population as clusters of non-reproductive butterflies at coastal sites in California has been known for more than 160 years. The earliest report of monarchs overwintering in California (Pacific Grove) dates back to 1864 [152]. They may have been present earlier, but surprisingly, they are not mentioned in any writings published during the Mexican and Spanish colonial periods. The first scientific documentation of monarchs overwintering in California was in the 1930 book ‘Migration of Butterflies’ by C. B. Williams [153]. However, biological studies on overwintering populations were not conducted until the 1960s–1970s [10,11,153,154,155,156]. We do not precisely understand the site conditions needed by monarchs for establishing and maintaining overwintering populations. However, monarchs appear to choose sites based on microclimate [157]. Protection from strong winds appears to be particularly important, along with reduced exposure to sunlight, although even these factors can be poor predictors of cluster locations [158]. Some access to early morning and late afternoon sunlight is important to allow butterflies to forage for water and nectar and reform clusters [118]. Overwintering sites that were favorable in the past may not continue to be favorable as climate change increases the incidence and severity of winter storms. 

Sites used by overwintering monarchs in coastal California are assignable to one of three categories: (1) treed gullies or ravines; (2) discrete, small patches of woodland; and (3) slightly elevated patches of woodland [118,119]. Overwintering monarchs at gully/ravine sites (usually oriented east-west and close to the ocean) may fare worse during severe storms when on-shore winds could create a ‘wind-tunnel’ effect. In contrast, roosting butterflies at wooded sites may be better protected from on-shore winds. Currently, there are 375 overwintering sites listed on the WMTG database [https://www.westernmonarchcount.org/ accessed on 2 January 2024] and new sites are reported annually. Heterogeneity in overwintering sites encompassing the three (perhaps more) categories noted above should maximize adaptation by overwintering monarchs. Although we should be concerned about the loss of overwintering sites through development or changed environment, the high density of sites along the California coastline may allow some latitude. The loss of an overwintering site is unlikely to result in migrants being ‘homeless’. They will, instead, likely continue their flight until they reach an alternative site.

The recent increase in winter-breeding in California [3,68,69] necessitates consideration of the viability of the habitat that is supporting this. To date, winter-breeding has only been reported from urban areas, principally San Diego, Los Angeles, Santa Barbara, and San Francisco, although this may reflect a lack of studies in natural areas. Breeding in urban areas is largely supported by non-native milkweeds (*A. curassavica* L., *Gomphocarpus fruticosus* (L) W.T.Aiton., *G. physocarpus* E.May.) which largely remain green throughout winter. However, in southerly areas, native milkweeds (primarily *A. fascicularis*) are increasingly remaining green for most of the winter. Even in San Francisco, *A. fascicularis* can remain viable for caterpillars into January. Winter-breeding in the South Bay area of San Francisco was successful during January–May 2021, resulting in a new generation of adults in April–May [69]. It is likely that two generations of adults are possible during winter in San Francisco, with the first generation flying in December. Concern exists that winter-breeding could be a ‘catastrophe’ for the western monarch population, by preventing migration and ‘losing monarch butterflies from the interior west’ [3]. These authors suggest that winter-breeding populations could lose the ‘genetic tendency to migrate’. Such a concern ignores the physiology of the monarch, which is programmed to respond to environmental cues that determine whether migration occurs or not [74,78,80,83]. While it is possible that over time a breeding population might lose its propensity to migrate, this would not happen within a single winter. The rebound in the western monarch population that occurred in 2021 strongly suggests that winter-breeding in California was beneficial (or at least had no effect) on the overall western population. Additionally, a strong argument could be made that the presence of non-native milkweeds in urban areas and winter-breeding during 2020–2022 was instrumental to the rebound of the western monarch population during 2021–2022.

### 3.4. Natural Enemies

All organisms are subject to mortality from natural enemies and abiotic factors, and the monarch is no exception. Estimates of mortality during monarch development range from 90–100% and are usually around 98–99% [159,160,161,162]. Most mortality occurs during the egg and early larval stages, primarily from predation [160,161]. Ten arthropod taxa were observed feeding on monarch eggs in Michigan, primarily at night [163]. Hermann et al. [164] described the predation of monarch eggs and larvae by 34 taxa under laboratory conditions. The abundance and diversity of natural enemies preying on monarchs varies geographically, with models indicating that parts of the western US have greater numbers of monarch natural enemies [165]. Research in the mid-west has shown that disturbance of larval monarch habitat by mowing suppresses the numbers of predators, giving monarchs a boost in survival [146]. Establishing a very diverse arthropod community on milkweed, including non-predators of monarchs, can also increase the survival rates of monarch eggs and larvae [166]. A small increase in the survival rates of developing monarchs may have large ramifications for overall monarch population dynamics [167].

#### 3.4.1. Predators

Predators are the most diverse and abundant group of natural enemies that impact the survival of both immature and adult monarchs. While chemical defense accrued during larval development on milkweed, helps adult monarchs by limiting attacks from vertebrates like birds [168], this defense rarely extends to invertebrate predators. Predators from at least nine orders and 17 families of insects and spiders can feed on monarch eggs and/or small larvae [164], and this is likely an underestimate. The most common predators of monarch eggs and larvae include ants, spiders, true bugs, ladybeetles, lacewings, and predatory wasps. Monarchs in urban habitats are frequently subject to extreme predation pressure by the introduced European paper wasp, *Polistes dominula* (Christ, 1791) [169]. This invasive wasp can develop large populations in urban areas, removing the vast majority of caterpillars (not just monarchs) from entire neighborhoods. In New Zealand, *P. dominula* reduced monarch densities by 66% at an urban site [170]. Outbreaks of *P. dominula* decimating monarch populations have occurred in San Francisco and eastern Washington cities and towns [James unpubl. obs.], and likely occur occasionally in all urban areas of the west. Outbreaks of *P. dominula* appear to last for two to three years, and then subside. Populations of this wasp do not appear to reach high levels in natural habitats, and their impact on monarch populations in these habitats may be much less [170]. 

#### 3.4.2. Parasitoids

The most common group of monarch parasitoids are flies in the family Tachinidae. At least seven species of tachinids are reported from monarchs in the United States, with all present in the west [171]. Parasitism of monarch larvae by tachinids varies geographically and seasonally, but in areas where the flies are common, a rate of 17–20% is average [171]. Little is known of the incidence and abundance of tachinid flies that parasitize monarchs in the west, but they appear most common in southern California, particularly in urban areas. Parasitism by tachinids is common in the San Francisco Bay area during the summer but declines or is absent during the winter [172]. Tachinid parasitism also occurs in northern California and southern Oregon but is rare in Washington and Idaho (James, unpubl. obs.). Parasitic wasps are common natural enemies of many butterflies and moths, but few species have been recorded parasitizing monarchs in North America. *Pteromalus cassotis* Walker 1847 is a chalcid wasp that appears to be a specialist, parasitizing monarch pupae in parts of the eastern US [173]. To date, there are no records of this parasitoid in the west. *Trichogramma* spp. wasps are well-known egg parasitoids of Lepidoptera, but reports of parasitism of monarch eggs are rare [166] and they have not been recorded parasitizing monarchs in the western US.

#### 3.4.3. Viral, Bacterial, and Unidentified Pathogens

Viral and bacterial pathogens infect caterpillar populations of butterflies and moths [174]. When viral or bacterial epizootics occur in communal caterpillar populations, the impact can be obvious with larvae hanging from branches as bags of foul-smelling liquid [175]. The incidence of viral and bacterial pathogens in monarch populations is unknown but is likely low in the arid, hot landscapes of the west. However, in artificial environments, for example, in rearing containers/cages, the incidence is much higher because of unnaturally high densities of caterpillars, poor sanitation, a lack of airflow, and higher relative humidity. A polyhedrosis virus, sometimes known as ‘the black death’ frequently causes disease in monarch larvae reared under crowded conditions [176]. 

#### 3.4.4. *Ophryocystis elektroscirrha* (OE)

Monarchs are parasitized by the protozoan, *Ophryocystis elektroscirrha* McLaughlin and Myers 1970 (OE). Since its discovery in 1970, much research has been conducted on the impact of this natural pathogen on monarch biology and ecology. The majority of these studies have been conducted on eastern North American monarch populations. Major conclusions are that heavily infected individuals have higher mortality during development, are smaller at eclosion, are more prone to unsuccessful eclosion, live shorter lives as adults, and have lower fecundity [177,178]. Infected monarchs also appear to fly shorter distances and at slower speeds than healthy individuals [179]. A limited number of studies have been conducted on OE in western monarchs. In 1992, Leong et al. [180] showed that 53–68% of monarchs at two overwintering sites were infected with OE, while in 2000, Altizer et al. [181] reported a 30% infection. Satterfield et al. [182] reported 8% prevalence at overwintering sites, but 74% at year-round breeding sites in southern California. A similar level of infection (69.3–77.5%) was found in winter-breeding monarchs in the San Francisco Bay area [69]. OE infection in western monarchs is more virulent than in eastern monarchs [178,183]. A recent study on OE in monarchs sampled over 40 years indicated a four-fold greater level of infection in the west than in the east [184]. However, there are no studies on the geographical or seasonal incidence of OE in the west. No studies have looked at the impact of OE on monarch biology and ecology under natural conditions in the west, except for a brief opportunistic study comparing the viability of OE-infected and uninfected migrant monarchs. Although only a small dataset was available, there was no difference in the distance traveled or apparent longevity of monarchs despite high levels of OE infection [66]. More research on OE and its impact on western monarch ecology, particularly in its summer-breeding range, is needed to understand its importance in the western US.

### 3.5. Non-Native Milkweeds

Three non-native milkweed species are available from plant nurseries in western North America. The most widely available species is *A. curassavica* (Tropical Milkweed, Blood Flower) is native to the American tropics and most likely an ancestral host plant for monarchs. The other two species are African milkweeds, *G. fruticosus* (Swan Plant, Narrow-leaved Cotton Bush), and *G. physocarpus* (Balloon plant, Hairy Balls), which are very similar and hard to separate. All three species are attractive garden plants with red, orange, or white flowers, contributing to their popularity among the general public. 

Non-native milkweeds are considered by some to have an adverse impact on monarch populations. Opponents of non-native milkweeds cite two reasons why they are not compatible with healthy monarch populations; enhanced ability to disseminate OE infection [185] and interference with monarch migration and reproductive dormancy [186]. The ability of non-native milkweeds to sustain OE infections in monarch populations stems from their persistence as green plants during winter in much of coastal California. In contrast, most native milkweeds in California have historically died back in the autumn until new growth in the spring. This creates a break in the OE infection cycle, which continues unabated on non-native milkweeds. However, as the climate warms in California, some native milkweeds are persisting longer into the winter as green plants and will also serve as reservoirs of OE. Pruning back non-native milkweeds in the autumn is recommended as a way of minimizing the survival of OE during the winter (https://xerces.org/blog/tropical-milkweed-a-no-grow. accessed on 2 January 2024). However, with monarch winter-breeding becoming increasingly common in California, the cut stems attract egg-laying females and can result in egg-dumping. In addition, milkweed with monarch eggs and larvae may be pruned and composted (James, unpubl. obs.). While autumn-pruning of non-native milkweeds originally had value for OE management, it is now detrimental to contemporary winter-breeding monarch populations in many urban areas of California and needs reconsideration. 

Major interference by non-native milkweeds in the induction and maintenance of monarch migration and reproductive dormancy is unlikely. The notion that the presence of healthy milkweed plants is capable of determining migration and reproduction comes primarily from conjecture and limited scientific evidence [186]. The role of host plants in determining the induction and maintenance of migration and reproductive dormancy in monarchs appears to be secondary to the primary cues of declining day length and fluctuating, decreasing temperatures [74]. Induction of reproductive dormancy in Australian monarchs results from exposure to cool temperatures (6–15 °C) for 2–4 days post-eclosion [187]. In Australia, monarchs become migratory and remain non-breeding despite the presence of non-native milkweeds (*G. fruticosus*, *G. physocarpus*), their major host plants in that country [73]. Furthermore, overwintering colonies only occur at sites that have milkweed present [72,188]. The presence of milkweed at overwintering sites has no apparent detrimental effect on the persistence of overwintering populations and the maintenance of reproductive dormancy [188].

Overwintering sites in New South Wales, Australia, are located 25–50 km from the coast [186], which differs from overwintering sites in California, which are predominantly within one kilometer of the coast (https://www.westernmonarchcount.org/map-of-overwintering-sites/ accessed on 2 January 2024). Native milkweeds in western North America do not thrive within two kilometers of the coast, so they are not present at overwintering sites. However, judging from the Australian situation, it is unlikely that if milkweed did occur at overwintering sites, it would interfere with the non-breeding status of monarchs. The presence of milkweed, native or non-native, two to five km inland from overwintering sites would likely be beneficial to post-overwintering monarch reproduction. 

Currently, four counties in California have banned the sale of Tropical Milkweed in nurseries, although the two other non-native milkweeds may still be sold. Tropical Milkweed is a common garden plant in California, and banning it from sale in some counties will not have any impact on monarchs. Any further restrictions on the availability and/or cultivation of Tropical Milkweed in California need to carefully consider the service that these plants provide for winter-breeding populations of monarchs. Winter-breeding populations studied in the South Bay area in 2021 were largely supported by the cultivation of non-native milkweed [69]. It is unlikely that these populations would have been as productive in the absence of non-native milkweeds. It is possible that the rebound in western monarch populations during 2021–2022 was in part fueled by the availability of non-native milkweeds for the development and productivity of breeding populations during winter 2020–2021. Naturally, the impact of any exotic species like *A. curassavica* should always include consideration of all biota within the invaded ecosystem, not just a single species.

### 3.6. Human Interference

Of the six stressors on western monarch populations described here, direct human interference through collecting and/or rearing monarchs is likely to have the least impact on populations. Humans indirectly and directly interfere with monarch survival in a major way by spraying pesticides, destroying habitats, and influencing climate change, as discussed above. However, put into perspective, human collecting and rearing of monarchs affects only a tiny portion of the population compared to the major human impacts cited above. The collection of monarchs for insect collections for some entomology education courses, as well as by amateur lepidopterists, does occur but is minimal. Similarly, the collection of monarchs for scientific research is limited and often regulated (e.g., https://nrm.dfg.ca.gov/FileHandler.ashx?DocumentID=194943&inline accessed on 2 January 2024).

As an iconic butterfly, monarchs have been reared by children and people interested in nature for many decades, wherever they occur [189,190,191]. Typically, a few monarch caterpillars are found in the backyard on milkweed and reared indoors. Like the rearing of tadpoles into frogs, the rearing of monarch caterpillars into butterflies has been an important part of childhood in North America for at least a century. Important, because these simple, childhood experiences of metamorphosis, create an indelible legacy of love and appreciation for wildlife. Research has shown that children with experiences of nature during childhood, like rearing monarchs or other butterflies, go on to be adult ambassadors for wildlife and conservation [192,193,194,195]. The importance of nature experience is recognized by schools that commonly use butterfly-rearing programs as part of science curricula. Today, most school butterfly rearing programs in the western US use Painted Lady butterflies (*Vanessa cardui* L.), but monarchs were also used in the past.

The public interest in butterfly rearing has been exploited by a number of commercial organizations that rear and sell butterfly livestock for private rearing. Thus, ‘butterfly farms’, or insectaries today in the eastern US supply monarch butterflies either as larvae for rearing or as adults for release at ceremonies like funerals and weddings. USDA permits are required for shipping monarchs between states and no shipment of monarchs is allowed across the Continental Divide. Currently, monarchs are not available commercially from butterfly farms in western North America, primarily because of current California state regulations prohibiting the handling of monarchs (see below). Commercially-reared monarchs may differ genetically and morphologically from wild monarchs and may not show correct migration behavior [196,197].

Home rearing was recently added to the growing list of anthropogenic activities considered to cause harm to monarchs. Since 2019, two scientific articles purporting to show that captive home-reared monarchs are less able to migrate and survive overwintering than wild monarchs were published [196,197]. An additional study showed that captive-reared monarchs may be weaker, paler, and have shorter forewings than wild migrants [198]. However, the Tenger-Trolander studies [196,197] were flawed by their use of static instead of declining day lengths in attempts to produce migratory adults in a wild population [74,199]. They also imperfectly examined migration induction in adult monarchs by insufficient post-eclosion exposure to inductive conditions [187]. These studies also used a flight simulator post-eclosion, rather than post-release tracking to assess migratory behavior of captive-reared adults. This was rectified in a later study by Wilcox et al. [200] where captive-reared monarchs were tracked as adults outdoors (rather than on a flight simulator), and these butterflies showed natural orientation southwards. These experiments were not perfect [201], but do indicate that future experiments on the effect of rearing conditions on adult flight behavior should have some natural outdoor evaluation component. The Tenger-Trolander studies received substantial media coverage and led to recommendations from conservation organizations against captive rearing. Evidence from tagging studies in the eastern US suggests captive-reared monarchs are less successful in reaching the Mexican overwintering grounds than wild-tagged monarchs [202]. However, the opposite is true for the western population. From 27,818 captive-reared monarchs in the Pacific Northwest during 2012–2019, there were 182 (0.65%) recoveries in California. In contrast, only one (0.075%) of 1325 wild-tagged monarchs was recovered [66,203]. The reasons for this discrepancy between eastern and western populations are unclear but are perhaps related to the shorter distances traveled by western monarchs to reach overwintering sites.

In conclusion, examination of the scientific evidence to date does not indicate that home captive-rearing of wild monarchs is overly harmful to western monarch populations, provided they are reared under strict hygienic conditions. To avoid disease problems, only small numbers should be reared and just one generation annually. Rearing containers/cages should be kept clean of frass and frequently sterilized with a bleach solution [189]. Even though captive-reared adult monarchs will likely determine their migratory/reproductive status after release via post-eclosion environmental cues [187], it is best to rear them under as natural conditions as possible, particularly with regard to exposure to natural daylengths. There are many online guides to rearing monarchs responsibly and successfully (e.g., https://monarchwatch.org/rear/ accessed on 2 January 2024) and a number of books [189].

#### The Value of Rearing Monarchs

There are two main benefits from rearing monarchs: aiding scientific research and developing in people a respect for the value of monarchs, which leads to a general appreciation of insects and nature. From basic science on metamorphosis and insect structure [204,205], to the latest genomic studies [206], the monarch is a favorite model of research in diverse disciplines. Monarch rearing by community and citizen scientists was fundamental to identifying the biogeography and dynamics of monarch migration in the Pacific Northwest [66,203]. Contemporary research on western monarchs aimed at determining appropriate actions for conservation invariably includes components centered on laboratory rearing [207,208].

Arguably, the greatest value of rearing monarchs is the long-term impact it has on the person doing the rearing. An example of this is the impact that rearing monarchs had on incarcerated men as part of a mental health program at the Washington State Penitentiary during 2012–2018 [66,203,209,210]. These inmates, many of whom were imprisoned for life, showed remarkable care and skill in caterpillar rearing. In addition to improving mental health, rearing monarchs instilled a realization that change is possible. For many children in North America and Australia, rearing monarchs is their first and pivotal experience of the natural world, engendering an appreciation for nature that can lead to a life-long commitment to nature conservation [211]. The theft of the joy of nature risks the apathy of future generations to its conservation [195]. 

Currently, the State of California does not allow anyone to handle monarchs without a permit. This means that a child cannot experience that first interaction with nature that the monarch provides. While rearing other butterfly species like Painted Ladies can also provide an important nature experience for children, the magic and charisma of rearing experiences with the monarch are arguably in a realm of their own. Robert M. Pyle in 1978 first coined the term ‘extinction of experience’ to describe what he saw as a worrying erosion of the ability of children to experience nature as they grew [212]. The extinction of experience is now considered to be a key environmental concept [213], that inhibits and undermines support for pro-biodiversity policies. With an insect biodiversity and abundance crisis already here [214], support for pro-biodiversity policies has never been more important. The long-term interest and commitment to monarch and nature conservation engendered by citizen personal experiences, far outweigh any perceived short-term benefits to the species, gained by prohibition. 

## 4. Conclusions and Perspectives

This review presents a holistic perspective of the history of monarch populations in western North America and the threats and opportunities that will shape future populations. This ‘big-picture’ view of western monarchs, derived from published research and informed by four decades of monarch research by the author, presents an optimistic view of how monarchs are faring. Further, it forecasts a future for western monarchs that is rosy, because it puts greater credence in the ability of monarchs to adapt to changing circumstances. Most of the current alarms and warnings about the future of the monarch are based on a narrower perspective of some of the stressors the species faces. Many of the factors often highlighted as being threatening appear far less consequential when considered holistically. The monarch butterfly, as a threatened icon, has inevitably attracted divisive political attention both within and outside the scientific community. Effective conservation of the monarch into the future requires consideration of all the scientific evidence we have available without the distraction of views fashioned by politics. As Yang [215] noted, ‘the complexity of this system should inspire humility’.

## Figures and Tables

**Figure 1 insects-15-00040-f001:**
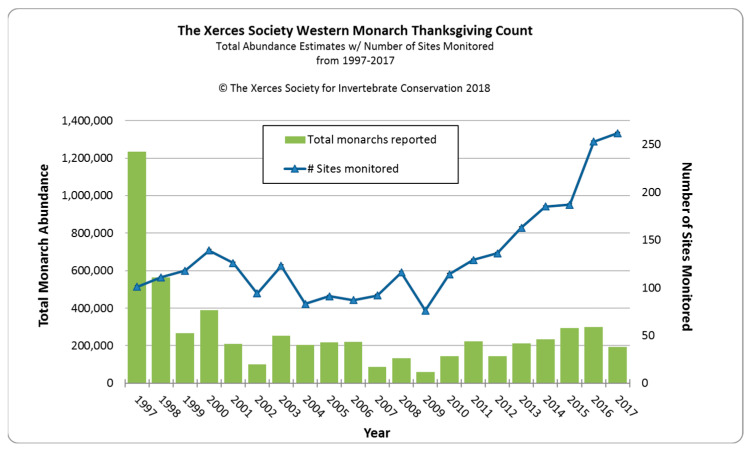
Western North American overwintering populations of monarch butterflies for 1997–2017 as assessed by annual Western Monarch Thanksgiving Counts (The Xerces Society for Invertebrate Conservation).

**Figure 2 insects-15-00040-f002:**
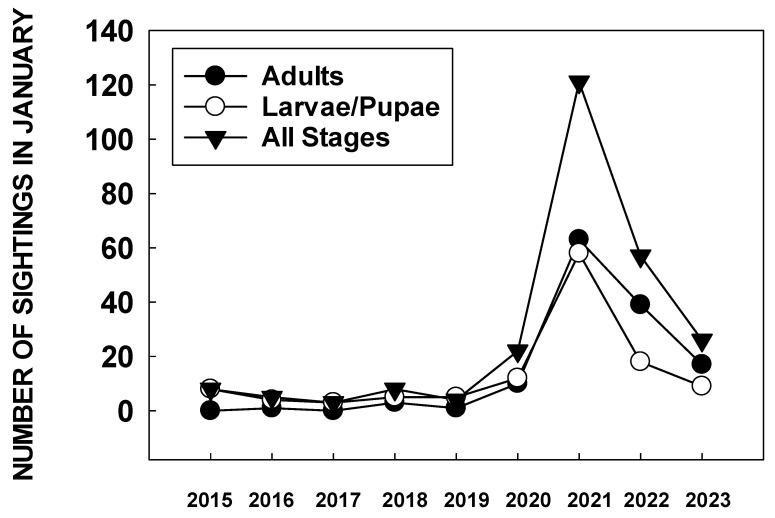
January (2015–2023) sightings of monarch larvae/pupae and adults in the bay area of San Francisco reported to https://www.inaturalist.org./ accessed on August 19 2023.

**Figure 3 insects-15-00040-f003:**
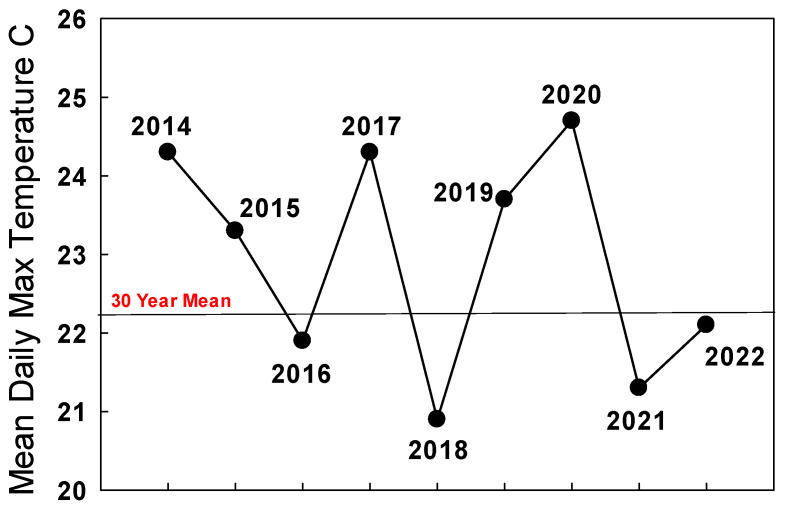
Mean daily maximum temperatures (°C) for San Francisco for September–October 2014–2022. Thirty-year mean: 1985–2015.

**Table 1 insects-15-00040-t001:** Thanksgiving count estimates of monarch butterflies at overwintering sites in California for 2018–2022 (Xerces Society for Invertebrate Conservation).

Year	Total Monarchs Reported	Number of Sites
2018	27,721	213
2019	29,436	242
2020	1899	249
2021	247,246	284
2022	335,479	272

## Data Availability

The data presented in this study are available in the article.

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
