# Peer review of "Monarch Butterflies in Western North America: A Holistic Review of Population Trends, Ecology, Stressors, Resilience and Adaptation"

_insects, 2024, doi:10.3390/insects15010040_

Round 1

Reviewer 1 Report (Previous Reviewer 2)

Comments and Suggestions for Authors

I appreciated that the ms was updated with some recent data as I requested (Fig. 2). This provides further support for the "breeding elsewhere" hypothesis. The new Table 1 of overwinter numbers from 2018-2022 makes it easier for the reader to follow the narrative.

In general, this is a thorough, comprehensive and well-written analysis of the population ecology of the western monarch population.

minor edits:

line 103 put "100,00-300,000" to be clear

line 109 comma before "primary"

In Table 1, Figs. 1,2 and 3 you use "during" when "for" is a better word

Same thing in Fig. 5 where "in", rather than "during" is better

Author Response

Thank you for your comments and suggestions and your approval of the revised Figure 2 and new Table 1, making it easier for the reader to follow the narrative. I have included all of your minor edits in the revised version.

Reviewer 2 Report (New Reviewer)

Comments and Suggestions for Authors

Author Response

Thank you for your very detailed and comprehensive review of my paper. I am pleased that you see the need as I do for a timely review of western monarchs. 

I understand and have complied with your request that I shorten the paper substantially by removing extraneous verbiage and unnecessary dialogue, as well as superfluous figures. The text part of the paper now runs to 21 pages instead of the previous 28.

I have also taken note of the need to remove unsupported information or to provide sources of information that may relate to situations with the western monarch (eg the storm issue and data from Mexico). I have done my best to de-emphasize my extrapolations and rely more on the literature.

I agree with and have altered the ms to reflect all of your specific comments except for:

Lines 243-254: I accept much of this section is speculative and have now altered the text to include words like 'likely', 'possible' and 'may' to emphasize the speculative nature. I have also reworded the text to leave open the possibility of summer declines. As you guessed there are no data on these matters.

Lines 269-270: The expected number of tag recoveries (0.8%) were made relatively close to but not at the overwintering sites. That is, they had virtually completed their migration (50-100 km from coast), so major mortality during the migration seems unlikely.

Lines 299-311: Unlike the Florida situation, the focus in this section is on reproductive status not phenotypic plasticity per se. So I dont think adding the Florida literature is necessary or appropriate. 

Line 334: The reference (68) is cited here as the source of the weather data, because I'm referring to this paper and what it presents. 

Line 393: We dont know how temporary these 'hot weather' roosts are. Certainly, in Australia they persisted for weeks... so the butterflies had dropped out of the migration. I'm assuming the same for the Texas roosts but we dont know. Taylor thinks these roosts do persist for more than a night or two.

Line 613: These early season milkweeds are mostly found in higher elevation areas, so will be less advanced than coastal monarchs. Text now added to reflect this.

Lines 750-760: This is in section 3.3 Habitat as a stressor, so I think its appropriate to have this info in this section as fire within a habitat may be a stressor.

Lines 901-919: Trichogramma wasps have not been recorded parasitizing monarch eggs in the west as far as I can tell. However, I have now included mention of eastern monarch parasitism by Trichogramma.

Lines 1070-1079: I prefer to keep mention of the practice of home-rearing garden-found caterpillars separate from mention of commercial supply of monarchs. These are 2 very different things, the first with minimal risk, the second with major risks.

My affirmative responses to your specific comments are shown below

Specific comments:

Line 34 Need a comma after “consistent”.  Ok

Lines 47 to 50. Do not use the tilde when referring to approximations. The author needs to simply used a word like “about” or “around” or at the very least the abbreviation “ca.” ok

Line 47. 200-400,000 needs to be changed to 200,000-400,000 unless the author really means that the lower end of the population estimate was just 200 individuals, which seems unlikely. ok

Line 87. The word “This” should be replaced with the word “They”. Ok

Line 102. 200-300,000 needs to be changed to 200,000-300,000 unless the author really means that the lower end of the population estimate was just 200 individuals, which seems unlikely. ok

Why is the top end of these values different than that cited on line 47? Corrected

Line 103. 100-300,000 needs to be changed to 100,000-300,000. Also change “fluctuated within this narrow 100-300,000 range” to “fluctuated within a range of 100,000-300,000 individuals”. I am not sure I would characterize 100,000 to 300,000 as a “narrow” range. Ok Corrected

Lines 117 and 118. The figure caption needs a formal citation. Ok

Lines 228-229. The numbers here are inconsistent with the numbers in lines 102 and 103. Ok corrected

Line 240. Please find a more authoritative source than Wikipedia. Try using the NWS database through NOAA. Ok

Lines 243-254. Much of this discussion seems either highly speculative or correlative. The author needs to either back up the information with published information or they need to point out the short-comings associated with extrapolating causations from correlations. Population changes between 2016 and 2017 do not necessarily reflect what happened in the winter of 2016. The author cites an 86% decline from January-March 2018 to 30,000 in 2018. When was that second census in 2018 conducted? If that second census was the Thanksgiving census, then it does not provide strong support for the idea that winter storms caused the decline. It could have been due to any form of die-off or reproductive failure that occurred between March 2018 and Thanksgiving of 2018. Was any data ever collected on die-offs associated with winter storms (ie. Observations of large numbers of dead butterflies)? To attribute declines to winter weather there would need to be data on corresponding declines in the number of reproductive individuals in the spring and summer of years following the winter storms. Does this data exist? “Winter storms” is a very vague statement. I realize that you describe those at lines 575-575, but it would be useful to briefly describe them here. Lastly, is there any data to support the contention that early dispersal limits female reproductive potential? I accept much of this is speculative but have used words like ‘likely’, ‘possible’ and ‘may’ to indicate this speculation. I have reworded the text to leave open the possibility of summer population declines. There are no data on all of these matters.

The data in Table 1 needs some form of citation. Ok

Line 265. Find a word other than “heyday”. Ok

Line 268. WSU needs to be defined. Ok

Lines 269-270. What is the recovery rate of marked individuals in years with high populations. For eastern monarchs the recovery rate is between 0.8% and 0.9%. At that rate, one might only expect to recover 11 individuals. Zero might then be within the range of statistical possibility. Also, the lack of recoveries might support an alternate hypothesis, namely that the decline observed in the 2020 Thanksgiving counts was due to mortality during fall migration. If that were true, and the 93.6% decline was due to mortality during fall migration, is it surprising that no marked individuals were recovered? The ‘expected’ number of tag recoveries were made (0.8%) relatively close to but not at the overwintering sites. That is, they had virtually completed their migration except for the last 50-100 km. So it is unlikely that there was major mortality during migration.

Lines 277-278. Delete “perhaps the tip of the iceberg.” This verbiage is unnecessary and makes the sentence structure awkward. Ok

Line 290. “talented and enthusiastic” Was this measured in some quantitative way? If so, then provide the data. If not, then it is extraneous verbiage and needs to be deleted. ‘ok’

Line 299. “and tellingly” Delete. Ok

Lines 299-311. There is considerable data on shifting phenotypes in Florida monarchs. The author needs to include that information here. Importantly, in Florida this phenotypic plasticity is not associated with reciprocal changes in breeding vs non-breeding populations. The focus here is on changes in reproductive status not phenotypic plasticity

Line 307. “20-40,000”. Twenty seems like an awfully small number. Corrected

Lines 334-334. Please cite the source of the weather data. Here and throughout the paper The reference (68) is cited as the source of the weather data presented here, because I am referring to this paper and what it presents.

Line 379. “Like a phoenix rising from ashes” Delete – please avoid use of common tongue and extraneous jargon throughout the paper. Okay

Line 393. “roosts in hot autumn conditions in Texas”. Delete this. These are temporary roosts of migrant individuals, not overwintering populations. There is a small, resident population along the Texas Gulf coast, but they do not form roosts. We don’t know how temporary these roosts are. They may be intermediate between overnighters and overwinterers. O R Taylor is a respected/trusted source and my Aust ref shows that ‘hot roosts’ can occur for weeks.

Line 408. Why the two numbers? This is confusing and needs to be clarified. ok

Line 409. “winter breeding populations in the San Francisco area were 409 again limited (Fig. 3).” The way this is written makes it sound like figure 3 contains data on the winter breeding population. But figure 3 is weather data. Ok-removed

Lines 411 – 427. This entire paragraph could be reduced by at least half. Ok

Lines 460-463. This statement needs citations. Removed

Line 613. Not sure I follow this argument. Above average temperatures can also advance the growth of milkweed plants. Can this statement be corroborated with quantitative data? If not, then it needs to be qualified with the possibility that there is no temporal displacement in warmer years.  These milkweeds mainly occur in mid-high elevations thus will not develop as fast as coastal monarchs. Text added to reflect this.

Figure 4. One monarch dead on the road is not a very convincing image of “conspicuous” mortality. The figure is, therefore, not necessary.  Ok

Lines 659-661. A citation is needed here. Ok

Lines 679-681. “Milkweed declines in California, the center of western monarch production, have likely been greater, but unlike eastern North America [129,130] there are no published data on this.” What is the basis of this statement? If there is no data, how do you know that the declines in California have been greater? The problem is how this sentence is worded. Re-write the sentence so that the uncertainty is much clearer. Ok

Lines 750-760. This information is redundant to that presented earlier under “stressors”. Incorporate this information the section on stressors and remove it from here.  This is in section 3.3: (Habitat as a stressor), so its entirely appropriate to have this information in this section. Fire within habitat may be a stressor

Lines 813-820. Citations? Quantitative data? The statement “Overwintering monarchs at gully/ravine sites (usually oriented east-west and close to the ocean) appear to fare worse during severe storms” implies that this has been quantified. If not, then this entire section needs to be shortened to include only the published literature on the subject. Ok modifications made

Lines 850-851. There are papers on eastern monarchs that support this idea. They should be mentioned and cited here. Ok

Lines 902-904. Delete the kitchy verbiage. In fact, delete these three lines entirely. Ok

Lines 901-919. Trichogramma infections can be an important source of mortality of monarch eggs. That needs to be mentioned here. Trichogramma wasp parasitoids have not been recorded parasitizing monarch eggs in the western US. Mention now made of Trichogramma as a parasitoid of E US monarchs.

Line 983. If the report is verified, then it needs a citation. Otherwise it is not verified. Ok

Lines 1023-1040. I fully agree that the negative impacts of A. curassavica on monarchs have been overemphasized to the point of irrational false panic. However, while I do not know the specific rationale used by the county governments in California for banning exotic milkweeds, I do think that there are risks in the use of exotic species that might have broader ecological implications. A. curassavica has now become naturalized in several California counties and, owing to the airborne seeds, has the capacity of expanding considerably further thereby making it a potentially invasive species. The authors need to include the counterpoint that there are also potential negative ecological impacts from the use of exotic species in monarch conservation. At the most extreme, saving one species could have cascading impacts on entire ecological communities of other species. Agreed. Cautionary sentence now added

Lines 1046-1054. While I agree with these points, they need some data to back them up. Is there any information on just how many butterflies are collected or how many are reared? In the absence of data, I think these two paragraphs should be combined and reduced to no more than a couple of sentences, especially since this topic is covered again in lines 1070-1079. Agreed

Figures 10 and 11 are unnecessary and should be deleted. Done

Lines 1127-1183. This section is extremely wordy and could be edited down to at least half the Done

Thank you again for your thoughtful review!

Reviewer 3 Report (New Reviewer)

Comments and Suggestions for Authors

This paper is ostensibly a review ranging widely on population trends, ecology, stressors, resilience and adaptation (to stressors) of monarch butterflies in western North America.  The author offers an interesting perspective colored in part by his experience of monarchs in Australia and more recently in the Pacific Northwest.  Although the claim is made that the review is holistic and comprehensive it is in parts selective and James (the author) centric.  At times I wondered if it was indeed a review or an opinion piece and critique.  I have no problem with the latter but in many places the narrative rambles and repeats.  How many times can one say, “more work is needed”?  This is a throwaway line.  A stronger review would specify what work needs to be done or question that need to be addressed.

I believe with major revision, shortening and tightening of the narrative the manuscript would be a useful contribution.  I have annotated the paper extensively that I hope gives an indication of what needs to be done. Alas I did not have the time to edit more extensively.  I hope the author takes the challenge on board. 

There are papers by Freedman, other than the Pocius paper which the author uses solely to justify rearing, that would appear to be relevant, and I have listed some here:

Hemstrom, Freedman and others (2022). Population genetics of a recent range expansion in monarch butterflies. Molecular Ecology 31, 4544-4557.

Freedman et al. (2021). Are eastern and western monarch butterflies distinct populations? A review of evidence for ecological, phenotypic, and genetic differentiation and implications for conservation. Conservation Science and Practice, 3:e432.

Freedman et al. (2020). Two centuries of monarch butterfly collections reveal contrasting effects of range expansion and migration loss on wing traits. Proceedings of the National Academy of Sciences, USA 117, 28887-28893. doi.org/10.1073/pnas.2001283117. 

page2image1273377808

Freedman et al. (2020). Host plant adaptation during contemporary range expansion in the monarch butterfly. Evolution 74, 377-391. doi.org/10.1111/evo.13914.

Freedman & Dingle (2018). Wing morphology in migratory North American monarchs: characterizing sources of variation and understanding changes through time. Animal Migration 5, 61-73. doi.org/10.1515/ami-2018-0003 

Other papers by Dingle appear to be relevant on migration in western north America e.g. Dingle et al. 2005. Distribution of the monarch butterfly, Danaus plexippus in western North America. Biological Journal of the Linnean Society 85: 491-500

Incidentally, the Pocius paper is not just about rearing but calls for experiments which is what the author seems to be doing.

One of the challenges for monarch researchers is to tackle the ecology of the species at an appropriate spatial scale; work in a few patches in a county will no longer cut it.  There is a need to take a broader perspective on milkweed distribution and how it is utilised by females when laying eggs as they traverse the landscape.  “Patches” may be population sources or sinks depending on eggs invested and survival.  There has been some useful work on egg laying at a landscape scale by the group in Iowa e.g. Grant et al. 2018. Predicting Monarch Butterfly (Danaus plexippus) Movement and Egg-Laying with a Spatially-Explicit Agent-Based Model: The Role of Monarch Perceptual Range and Spatial Memory. Ecological Modelling 374: 37–50, that built on earlier related papers by a group in Australia, see:

Zalucki, M.P., Parry, H. & Zalucki, J.M. 2016. Movement and egg laying in Monarchs: To move or not to move, that is the equation. Austral Ecology 41: 154-167.

Zalucki, M.P. and Lammers, J. H. 2010. Dispersal and egg shortfall in Monarch butterflies:  what happens when the matrix is cleaned up? Ecological Entomology 35: 84-91

Again, these papers are interesting but need to be properly tested with appropriate data at large spatial scales.

Comments on the Quality of English Language

Author Response

Thank you for your thoughtful and useful comments.

Following your recommendations, I have shortened and tightened the ms, following and adopting all of the suggestions you made as annotations on the ms. The length of the body of the paper has been shortened from 28 to 22 pages. I have also included the suggested Freedman et al 2021 paper which as you suggest is clearly of relevance to the western population.

My specific responses to your annotations are shown in a separate PDF

Round 2

Reviewer 2 Report (New Reviewer)

Comments and Suggestions for Authors

First paragraph of introduction:  "200-400,000" needs to be changed to "200,000-400,000". Otherwise everything looks fine.  

Author Response

  1. The first paragraph of Introduction: 200-400,000 has now been changed to 200,000-400,000 as requested.

Reviewer 3 Report (New Reviewer)

Comments and Suggestions for Authors

I look forward to seeing the paper in print.

Author Response

Thank you. I too look forward to seeing the paper in print!

This manuscript is a resubmission of an earlier submission. The following is a list of the peer review reports and author responses from that submission.

Round 1

Reviewer 1 Report

Comments and Suggestions for Authors

In this manuscript, James aims to provide a background to the population dynamics of monarchs in western North America and to examine the relative importance of different stressors to population trends. Unfortunately, the review falls short on reaching these goals, as outlined below.

1. One of the major problems with this manuscript is that it is not a review, but a speculative opinion piece instead. The literature is cited selectively and misleadingly. There is no quantitative analysis or careful consideration of the relative contributions of different stressors. And there is a wealth of unfounded speculation. As a case in point, Figure 5 provides a word cloud in which font size corresponds to relative importance “as judged by the author”. The point of a review is to assess how existing literature provides evidence for different hypotheses, and to point out gaps in the literature. The point is not to use one’s own opinion to speculate about the relative importance of different factors, and then use some non-quantitative font size-based word cloud to visualize that speculation.

2. In the simple summary and abstract, the author’s speculations are portrayed as general truths. “The impact of natural enemies … is not seen as a major contributor in contemporary population fluctuations” and “Human interference (capture, rearing) has the least impact on monarch populations” are the author’s opinions, not conclusion based on careful consideration of existing science. Presenting personal opinions as scientific consensus is unscientific. The author follows the same approach in the rest of the paper. For example, in lines 1011-1015, the author states that human interference is considered to be the least problematic. Considered by the author, that is, not by many other scientists.

3. What is lacking from this review is a balanced summary of existing studies on population dynamics in western North America. Many studies have been conducted, and the consensus of these studies is that it is hard to pinpoint specific causes to the dynamics. I would recommend building a table in which different studies are listed, and their results are summarized in a systematic way. The fact that these studies cannot pinpoint specific causes is important, and points at the complexity of these ecological dynamics, many of which we simply do not understand yet. Pointing out gaps in knowledge would be useful and helpful. Rather than dismissing the idea that mass-rearing and releasing of butterflies is not harmful – and even beneficial – the author should point out what studies are needed to determine whether this is the case or not. The author writes at long length of the numbers of monarchs in different years, but a thorough review of the actual population modeling studies that have been done, is lacking. A systematic overview of what these different studies have suggested about the different stressors is also not here.

4. The idea of neonicotinoids being important is interesting, but speculative. While there are studies that have looked at the effects of these chemicals on monarchs (with mixed results, some indicating no detrimental effects), I am not aware of studies that have addressed their importance at a population level. This is another gap in the literature which would have been useful to point out.

5. The figures are not helpful, and are consistent with the lack of systematic review and quantitation. Only a few figure show data. The remainder act as anecdotes or serve to convey emotions rather than quantitative analysis. Figure 1 starts with numbers in 1997. The author mentions counts exist from before then. They should be shown. Why does the figure stop in 2017? This makes no sense given the heavy focus on monarchs declining further and then bouncing back in 2021. Figure 2 is not clear. Are the data in Figure 3 normalized for effort? Or could the increased numbers in 2020/2021 be due to the Covid pandemic, when many people started using iNaturalist? As mentioned, Figure 5 is completely unsubstantiated. The monarch in Figure 6 looks like it has been staged; a figure showing the number of road deaths or similar would be more helpful. Figures 12 and 13 seem inappropriate, showing prisoners and police officers, the latter captured unknowingly on a door camera (also, where can this information of this arrest be verified?).

6. The author lists several potential explanations for the monarch population bouncing back in 2021. While it is possible that winter breeding monarchs contributed to this, this has not been demonstrated (another gap in our knowledge that would be worth pointing out), it may also be possible that monarchs bounced back because of new influx from Mexico. Eastern and western monarchs are genetically indistinguishable, forming a single population (Talla et al. 2020), and it is possible that the western numbers are fed by eastern monarchs over time. Thus, the idea that winter-breeding monarchs are beneficial, or even needed, to support the perseverance of monarch migration in the west (e.g. lines 993-1009) is unsubstantiated at this time. The author should call for more research, rather than jump to conclusions. Moreover, it is not accurate that other scientists have sent the message that tropical milkweed is bad for monarchs (line 993): what scientists have messaged, based on the data at hand, is that tropical milkweed could interfere with migration and that it contributes to sky-rocketing parasite prevalence.

7. Lines 824-833. The author argues here that there is no reason that winter-breeding monarchs could lose the genetic tendency to migrate, and argues that migration is solely determined by physiology and environment. This is not the case. Genetic work has clearly shown that non-migratory monarch populations are different from migratory monarchs, and that they have differentiated genes (Zhan et al. 2011). True, we do not know this yet for newly formed winter-breeding monarchs in North America, but the idea that there would be no selective pressure to abandon migration simply makes no sense.

8. The author makes the point that because most work on Ophryocystis elektroscirrha has been done on eastern monarchs, this work does not provide meaningful insights to western monarchs. This makes no sense. Many studies have been done on the basic biology of this parasite and eastern and western monarchs. Given that eastern and western monarchs are genetically indistinguishable, and form a panmictic population (Talla et al. 2020), there is no reason to believe that findings in eastern or western monarchs should not translate to each other. The statement that “Only three studies have been conducted on OE in western monarchs” is completely wrong. In addition to the studies mentioned by the author, the following studies have specifically studied parasite prevalence in western North America, or conducted experiments with infection of western monarchs: (Altizer et al. 2000, de Roode et al. 2008, de Roode and Altizer 2010, Lefèvre et al. 2010, Lefèvre et al. 2011, Lefèvre et al. 2012, Sternberg et al. 2012, Satterfield et al. 2016)

9. The review cites a lot of unpublished observations from the author himself. This is problematic, as readers have no way of confirming this information. Many of these observations could be observation bias (e.g. people writing to author about monarchs breeding in their back yard; numbers of monarchs killed on the road).

10. The author states that the idea that healthy milkweeds can break monarch diapause and interfere with migration is mostly based on conjecture and supported by limited evidence (lines 960-963). However, the paper by Majewska et al that he cites here (ref 184) clearly shows this, and there is other work too. For example, Satterfield et al (Satterfield et al. 2018) show that migratory monarchs mix with resident monarchs at sites with exotic milkweed, and that this is associated with increased reproductive activity and parasite infection. Moreover, James states that host plants only play a secondary role in inducing diapause in monarchs, and cites a paper by Goehring and Oberhauser (ref 74) to support that claim. However, that paper concludes that day length, temperature and senescing host plants act additively to induce diapause. It does not quantify the relative importance of each factor.

11. In lines 969-972, the author claims that Gomphocarpus spp. are the only host plants available for monarchs in Australia. This is really puzzling, because A. curassavica occurs in Australia as well. Maybe it is not as common, but I have collected monarchs myself from A. curassavica plants in the areas around Brisbane and Sydney.

12. In lines 971-977, the author makes the case that because host plants in Australia and New Zealand do not seem to interfere with migration or reproductive diapause, the case can be made that things will be fine in western North America too. This is especially ironic, because the author has dismissed any finding from eastern North America to make inferences on western monarchs (even though these monarchs belong to one panmictic population (Lyons et al. 2012, Talla et al. 2020)), while at the same time drawing parallels with Australian and New Zealand monarchs, which are actually strongly genetically differentiated (Lyons et al. 2012, Pierce et al. 2014, Zhan et al. 2014). Given their differences in genetic make-up, western monarchs may (or may not; we do not know at the moment) respond to changes in different ways than their Australian and New Zealand counterparts.    

13. On multiple occasions, the author misrepresents or selectively cites specific publications to make his point. For example, the author cites Yang 2023 as saying that “the complexity of this system should inspire humility”. In that paper, Yang continues to describe three actions to address declines in western monarchs, the second of which says: “Second, recognizing the limits in our current understanding, we should follow the precautionary principle to minimize the risk of counterproductive action…. In practice, this may mean prioritizing efforts to better understand and facilitate existing mechanisms of ecological resilience and recovery over direct actions to manipulate or augment the population with less certain consequences.” Thus, the humility encouraged by Yang includes preservation of the natural ecology of monarchs over the rearing and releasing of monarchs, as pushed for by James. Another example also relates to James’ argument to justify and encourage the rearing and releasing monarchs. The author cites the paper by Hayward et al 2022 named “Intergenerational inequity: stealing the joy and benefits of nature from our children” to legitimize this practice. However, this paper says nothing about rearing species, and instead emphasizes the importance of maintaining natural and wild areas for people to experience nature. While I fully agree that early experiences with nature are important for children, there are better ways to do this than by rearing monarchs in artificial conditions, and then releasing them into the wild. As part of my won conservation work I encourage people to create pollinator gardens. I provide plants for free to individuals and schools. The joy of attracting a range of wild pollinators – not just monarchs – to newly recovered ecosystems is an experience that will instill an appreciation for nature. When I tell people that rearing monarchs in confined conditions with high disease risk is like capturing people on a cruise ship, infecting them with norovirus or Covid, and then sending them to the world to infect others, they much prefer to recreate habitat than to contribute to disease spread.

14. It is unethical to encourage people capturing and rearing monarchs. Most people who rear monarchs maintain them at high densities and practice bad hygienic practices. Many monarchs become infected, and people unwittingly release detrimental parasites into nature. Even if reared monarchs may be able to migrate (the evidence is mixed on this), this argument is moot, as people should simply not propagate infections and release infected butterflies into the wild. The author suggests that rearing is fine as long as people rear small numbers, use hygienic practices and only do one generation per year. Although this sounds nice, there is no evidence that this is actually beneficial. Moreover, anyone who has interacted with individuals who rear monarchs will know that this is just not feasible. Again, there are better ways. Let people build gardens instead. Encourage them to take part in butterfly counts. Let them realize there is more out there than one species of majestic butterfly.

15. It is unclear what the author tries to convey with the section in lines 1037-1044. It is actually great that people in California cannot legally order monarchs from commercial suppliers. Commercial breeders create heavily inbred monarchs, and often ship out monarchs with heavy parasite infections. It is bad practice.  

16. The whole section on the value of rearing monarchs, starting on page 26, is beyond the scope of the review, as described in the introduction (to provide a background to the population dynamics of monarchs in western North America and to examine the relative importance of different stressors to population trends). Moreover, it is riddled with inaccuracies and unsubstantiated claims. The idea that people need to rear monarchs to appreciate nature simply makes no sense. People love panda’s and rally behind nature conservation because of them. Yet, no one rears them. The idea that people should interfere with nature, spread disease and potentially interfere with migration is unsound. I fully agree that people immersing in nature is important and crucial to developing a respect for nature. But why does that have to be done by rearing a single species in plastic tubs under unnatural conditions? Why not build habitat and see experience nature through many different species, rather than one monarch butterfly? It is hard to see that the author may not have some conflict of interest here. Having worked with citizen science programs to rear and release tens of thousands of monarchs may bias the author toward his belief that this practice is not harmful, and may even be beneficial.

Cited references

Altizer, S. M., K. S. Oberhauser, and L. P. Brower. 2000. Associations between host migration and the prevalence of a protozoan parasite in natural populations of adult monarch butterflies. Ecological Entomology 25:125-139.

de Roode, J. C., and S. Altizer. 2010. Host-parasite genetic interactions and virulence-transmission relationships in natural populations of monarch butterflies. Evolution 64:502-514.

de Roode, J. C., A. J. Yates, and S. Altizer. 2008. Virulence-transmission trade-offs and population divergence in virulence in a naturally occurring butterfly parasite. Proceedings of the National Academy of Sciences of the United States of America 105:7489-7494.

Lefèvre, T., A. Chiang, M. Kelavkar, H. Li, J. Li, C. Lopez Fernandez de Castillejo, L. Oliver, Y. Potini, M. D. Hunter, and J. C. de Roode. 2012. Behavioural resistance against a protozoan parasite in the monarch butterfly. Journal of Animal Ecology 81:70-79.

Lefèvre, T., L. Oliver, M. D. Hunter, and J. C. de Roode. 2010. Evidence for trans-generational medication in nature. Ecology Letters 13:1485–1493.

Lefèvre, T., A. J. Williams, and J. C. de Roode. 2011. Genetic variation for resistance, but not tolerance, to a protozoan parasite in the monarch butterfly. Proceedings of the Royal Society B-Biological Sciences 278:751-759.

Lyons, J. I., A. A. Pierce, S. M. Barribeau, E. D. Sternberg, A. J. Mongue, and J. C. de Roode. 2012. Lack of genetic differentiation between monarch butterflies with divergent migration destinations. Molecular Ecology 21:3433-3444.

Pierce, A. A., M. P. Zalucki, M. Bangura, M. Udawatta, M. R. Kronforst, S. Altizer, J. Fernández Haeger, and J. C. de Roode. 2014. Serial founder effects and genetic differentiation during worldwide range expansion of monarch butterflies. Proceedings of the Royal Society B-Biological Sciences 281:20142230.

Satterfield, D. A., J. C. Maerz, M. D. Hunter, D. T. Flockhart, K. A. Hobson, D. R. Norris, H. Streit, J. C. de Roode, and S. Altizer. 2018. Migratory monarchs that encounter resident monarchs show life‐history differences and higher rates of parasite infection. Ecology Letters 21:1670-1680.

Satterfield, D. A., F. X. Villablanca, J. C. Maerz, and S. Altizer. 2016. Migratory monarchs wintering in California experience low infection risk compared to monarchs breeding year-round on non-native milkweed. Integrative and Comparative Biology:icw030.

Sternberg, E. D., T. Lefèvre, J. Li, C. Lopez Fernandez de Castillejo, H. Li, M. D. Hunter, and J. C. de Roode. 2012. Food plant-derived disease tolerance and resistance in a natural butterfly-plant-parasite interaction. Evolution 66:3367-3376.

Talla, V., A. A. Pierce, K. L. Adams, T. J. de Man, S. Nallu, F. X. Villablanca, M. R. Kronforst, and J. C. de Roode. 2020. Genomic evidence for gene flow between monarchs with divergent migratory phenotypes and flight performance. Molecular Ecology 29:2567-2582.

Zhan, S., C. Merlin, J. L. Boore, and S. M. Reppert. 2011. The monarch butterfly genome yields insight into long-distance migration. Cell 147:1171-1185.

Zhan, S., W. Zhang, K. Niitepõld, J. Hsu, J. F. Haeger, M. P. Zalucki, S. Altizer, J. C. de Roode, S. M. Reppert, and M. R. Kronforst. 2014. The genetics of monarch butterfly migration and warning colouration. Nature 514:317-321.

Author Response

Response to Reviewer 1  

Thank you for taking the time to thoroughly review my manuscript.

Reviewer Point 1. I respectfully disagree with the assertion that this manuscript is not a review! It undoubtedly is, with a large number (> 200) of cited publications comprehensively selected to provide a balanced overview of monarch ecology/biology in western North America. 

If you consider that I cited literature ‘selectively and misleadingly’, this was not my intention. Give me examples of important missing literature and I will gladly incorporate. I do not claim to have cited all publications (there are many that present similar information/data), but I did aim to cite from the entire spectrum of relevant research.

When I ‘speculate’ or ‘opine’, it is only based on the available data and information. This is fair within the framework of a review (as the other reviewers agreed). I think ‘Informed speculation’ is justifiable in a review such as this, particularly if the author has a credible track record of monarch research dating back to 1978. If there is any truly ‘unfounded’ speculation, I am happy to remove it.

The ‘word cloud’ figure is an arbitrary way of presenting a simplified visual of relative importance of stressors and it is probably an unnecessary figure, that I agree, could be construed as ‘oversimplified’ and too ‘speculative’. It does not add much to the content and has been deleted.

I accept your point that reviews should point out ‘gaps’ in the literature and have edited accordingly.

Reviewer Point 2. I have adjusted the wordings in the simple summary and abstract to not sound like ‘general truths’. The statements are now more circumspect. For example ‘likely to be’ instead of ‘considered to be’

Reviewer Point 3. I have taken the approach of comparing studies on the different potential stressors (habitat, climate, pesticides) in their own sections, rather than comparing the results of different studies in the west together. The reader gets the same information either way. Indeed, the fact that these studies do not arrive at the same conclusions, argues for their coverage under the separate headings of habitat, climate and pesticides.

Reviewer states: “Rather than dismissing the idea that mass-rearing and releasing butterflies is not harmful- and even beneficial……”

I do not mention mass-rearing of monarchs in the manuscript. Also, I did not simply ‘dismiss the idea’, I stated that ‘rigorous examination of the scientific evidence to date does not indicate that captive-rearing is harmful’. I cited studies from the east and west, with the latter showing no impact on migration success. I agree that more studies are needed and have now added text to this effect.

Reviewer Point 4. Neonicotinoids: I have presented most of the pertinent research conducted to date on monarchs with this group of pesticides, both negative in impact (cited refs 29, 30, 44) and positive (30, 31,45). Consequently, my limited speculation is justifiable. Of course, more research is needed and I pointed this out in lines 203-206, 207-208, and 219-221.

Reviewer Point 5. I believe the figures are helpful in conveying the message of the ms. Figures don’t have to show data in a review. Figure 1 accompanies the important post 1997 discussion of populations. Pre-1997 population levels are far less reliable ‘estimates’ rather than ‘counts’ (as stated in lines 91-92). I have amended lines 78-79 to emphasize this and also show the references where these data can be viewed (line 93).

Figure 1 stops in 2017 because this section of the ms deals only with this relatively stable population era. The subsequent drastic declines during 2018-2020 are dealt with in section 2.4. I have now included a table (Table 1) showing overwintering population data for 2018-2022.

Figure 3. Consultation with I-Naturalist admin provided the figure of ~7% in overall increase in observations on the platform during the pandemic. So, while this may have contributed to the increased number of observations of immature stages during 2020/21, it was not the whole story. I have now added January data for 2022 and 2023 to this figure, which shows a clear decline in numbers of larvae/pupae during these years, as would be expected with more ‘normal’ fall temperatures and larger populations at overwintering sites, commensurate with my hypothesis.

Figure 5, removed.

Figure 6 is NOT staged! This was taken during a period of spring migration along the Trinity River in northern California during May 2014. Numerous road-kill monarchs were evident at this time. I have witnesses.

Figures 12 and 13: I disagree that these images are inappropriate. Figure 12 has inmate faces fuzzed out to remove their identities. The California Dept of Fish and Wildlife officers (not police) shown in Fig 13 are not identifiable. This image is a graphic representation of the non-proportional response of regulatory agencies to a wildlife infraction, that I believe needs to be available for viewing. There was no ‘arrest’, it was delivery of a cease-and-desist letter (as explained in the figure caption). If the reviewer needs verification, I can include the cease-and-desist letter within the ms, but personally consider this a bit ‘over-the-top’.

Reviewer Point 6. All the potential explanations for the population rebound in 2021 are discussed (winter breeding, migrants from Mexico, cryptic overwintering in CA) and I do point out that we don’t know which explanation is correct. I propose the idea that winter-breeding monarchs in California may be important to the sustainability of western monarchs, because earlier work showed that winter breeding in San Francisco during winter 2020/21 was substantial (69). However, I have now added text calling for more research on this point.

Clearly, ‘interfering with migration’ and ‘contributing to sky-rocketing parasite prevalence’ are ‘Bad’ but I have amended my statement on Tropical Milkweed being ‘bad’ for monarchs, to ‘The message that Tropical Milkweed is not recommended for monarchs….’

Reviewer Point 7. I disagree. Long-established non-migratory and migratory populations may differ genetically, but apparently retain their ability to switch back and forth as shown by Freedman et al (2018). So, even if a winter breeding population during one winter became genetically modified (unlikely), this would not take away the ability to respond to physiology and environment. The data of Freedman et al (2018) indicate that even long-term non-migratory monarchs retain their ability to physiologically respond to appropriate environmental cues and become migratory.

Reviewer Point 8. Although Talla et al (2020) did show eastern and western monarchs to be members of a panmictic population, they also showed that E and W monarchs maintain migratory differences despite gene flow, with variation driven by environmentally induced differential gene expression. Why couldn’t similar environmentally-induced differential gene expression also affect monarch parasitism ecology differently in the east and west? I am simply advocating that more research be done on OE in the west, particularly in terms of natural population dynamics of monarchs and OE.  I have corrected my statement that only three studies have been conducted on western monarchs. I have now included citations to the two additional papers that studied parasite prevalence in the west (noting that I had already included two of the extra citations that the Reviewer suggested).

Reviewer Point 9. There are 4 unpublished observation citations within 28 pages of text. I do not consider this excessive or problematic set against 213 publication citations.

Reviewer Point 10. I disagree that the paper by Majewska et al (ref 184) clearly shows interference by milkweed in breaking diapause/migration. This paper lacks an important experimental treatment, and in my view, should not have been published. My words ‘limited scientific evidence’ are entirely appropriate (and diplomatic) in pointing out that the concept of milkweed interference is based on a single (flawed) paper. The other paper the reviewer refers to (Satterfield et al 2018), simply shows an association rather than causation.

As I point out “Major interference by ornamental non-native milkweeds to induction and maintenance of monarch migration and reproductive dormancy is unlikely”. I didn’t say ‘impossible’ but the evidence to date does not strongly support this idea.

When photoperiod, host plant and temperature were examined together, Goehring and Oberhauser (Ref 74) showed that host plant had no effect on diapause induction (page 679). Other experiments reported in this paper did show an effect of host plants, so I have now added the qualifier ‘likely’ to my statement on line 966.

My thoughts on diapause induction in monarchs should be considered credible since I have published extensively on this subject (Refs. 72, 73, 78, 81, 83, 185).

Reviewer Point 11. Asclepias currassavica while commonly supporting year-round reproductive monarch populations in Queensland, is a rare host plant (only found in gardens) for seasonal monarch populations in New South Wales, Victoria and South Australia. Thus, the vast majority of migratory monarchs in southern Australia are linked to Gomphocarpus milkweeds. I have changed the sentence on line 970 to reflect this.

Reviewer Point 12. In lines 971-977, I provide facts concerning the milkweed monarch dormancy situation in Australia and New Zealand. I do not make the case that ‘things will be fine in western Norh America too”. I let the reader draw their own conclusions.

Further, I have not dismissed all findings on eastern North American monarchs as having relevance to western populations. I treat them with the same questioning and reservation that I treat findings on Australian monarchs.

Reviewer Point 13. Everything in this paper that I tacitly encourage is supported by the available literature and information. Limited, hygienic rearing of monarchs is supported by many scientists and goes hand in hand with developing and preserving habitat. The reviewer appears not to appreciate the thrill and captivation engendered by the up-close and personal experience of witnessing monarch metamorphosis! Or just seeing a butterfly close up! We hold an annual “Music and Monarchs” day locally which usually features a few reared monarchs that people can view up close.  This excites people of all ages drawing huge crowds to the event. In recent years, we have been unable to have monarchs at the event and the crowds are smaller and query “where are the monarchs?”.

Certainly, the rearing experience should always go hand-in-hand with pollinator gardening etc. The synergy between the two is dynamite! Rearing a single generation annually under hygienic conditions is never confinement in a cruise ship! Embracing both a rearing experience and creating habitat, rather than one or the other, is actually the point of this paper: look at things holistically!

Reviewer Point 14. Again, the holistic point is missed! Rear AND restore habitat. I believe the majority of people who rear ARE responsible and do it the right way. It is up to us to educate people to do things correctly.

Reviewer Point 15. This section presents the facts as a review should. I agree commercial insectary rearing of monarchs is bad practice.

Reviewer Point 16. I disagree. The section on the value of rearing (a controversial subject) is very important to include (as recognized and stated by Reviewer 3), considering the public interest in this and pushback from some quarters. It is within the scope of the review in that it is a potential part of the ‘Human Interference’ stressor. The section is well-supported by references and practical experiences (e.g. mental health of prison inmates). The latter is an excellent example of the human benefits derived from rearing an insect and was considered important enough to have a chapter devoted to it in a book on Citizen Science (Ref 207).

Rearing an insect like a monarch can be considered a ‘fast-track’ or ‘gateway drug’ to instilling love for and appreciating nature. As mentioned in Points 13 and 14, doing both things (close contact rearing and habitat creation) will be more effective than one or the other. The monarch is often the ‘gateway’ insect that leads a person to study/conserve other butterflies/pollinators, so again, we need a holistic view.

Reviewer 2 Report

Comments and Suggestions for Authors

This paper does a thorough and comprehensive job of reviewing what is known about the ecology of the western Monarch butterfly population. It provides an excellent analysis of possible reasons why the population declined from its historical high and then crashed and then rebounded, defying expectations. The decline from the historical high is attributed primarily to chemicals, principally neonics. The author acknowledges that the evidence is basically that there is a correlation between the decline in the population and the increase in neonic use. It would be great to have a representative sample of milkweed habitats throughout the spring and summer breeding ranges to see what proportion of habitats might be exposed to neonics to gauge the seriousness of this stressor. In general, the explanation for why the overwintering population was so low in 2020 then rebounded seems reasonable; i.e. that the majority of the 2020 population was somewhere else. Part of that explanation is that it was warmer in 2020 than usual in coastal areas like San Francisco which induced the migrants to forgo overwintering and breed in that region. However, Fig. 4 shows the same warmer than normal conditions for Sept.-Oct. in San Francisco in 2014 and 2017 yet we don’t see an uptick of monarch sightings in SF in those years (Fig. 3). Why is that? Could you add the levels of sightings for January of 2022 to Fig. 3? Given near normal S-O temps in 2021 there should be little winter breeding and the number of sightings should go back down. Certainly, more data are needed on winter breeding. It would seem that most of this would occur in urban areas where tropical milkweed is planted rather than in more uninhabited areas where native milkweed is unlikely to be active even in a warm winter. People in urban areas need to be recruited to provide data.

A general consideration: Provide a graph or Table for overwintering counts for the years 2018-2022.

Specific comments by line number:

101 really just corn and soybeans

102 just say “spray”; they are not spraying from airplanes

228 “was” characterized

238 add “in 2018”

246 “mooted” -new word for me

283-285 move to the head of the next paragraph

308 exposure to what?

315 “were” terminated

364 Simple mathematics shows

374 see recent paper in Scientific Reports: Overwintering and breeding

patterns of monarch butterflies (Danaus plexippus) in coastal plain habitats of the southeastern USA

Michael R. Kendrick  & John W. McCord

382-384 Pacific Northwest monarch sightings were much higher in 2022 than 2021 but the overwintering population sizes were fairly similar

454 Bt

513 pesticides are not likely to affect all habitats with milkweeds. Many of the milkweed habitats in the Pacific Northwest would not be near agricultural areas where pesticides would be used

531 in what year?

588 lack/difficulty ???

647 “wander” awkward sentence

679 trees

700 chemical ecology?

733 extra (

736 more attractive because of younger leaves

780 “forage for nectar” – but overwintering monarchs do little if any nectar foraging. They may need to access water to metabolize their lipids

1138 “adapt and evolve” – think suggests an evolutionary process while the evidence seems to show a plasticity or adaptation in the colloquial sense

Comments on the Quality of English Language

I have pointed out a few things

Author Response

Response to Reviewer 2

Thank you for your thorough review and positive comments.

I am pleased that you consider my explanation of the reasons behind the long-term and drastic declines to be reasonable. I have now added sighting data for January 2022 and January 2023 to Fig. 3 (now Fig. 2) as requested. The number of sightings of immature monarchs in these years did indeed go down, indicating that breeding in the San Francisco area was reduced when autumnal temperatures were closer to average, as per my hypothesis. I thank you for making this suggestion and improvement to the manuscript. Additional text concerning this has also been added (lines 361-63 and 394).

I have now included a table showing the overwintering counts for 2018-22, as suggested.

Comments by Line number:

Line 101: modified to corn and soybeans only

Line 102: Removed ‘aerially’

Line 228: Change to ‘was’

Line 238: Added ‘in 2018’

Line 246: ‘mooted’ = to bring up for discussion (https://www.merriam-webster.com/dictionary/moot)

Line 283-285: Now moved to head of next para.

Line 308: Added ‘exposure to temperatures optimal for reproduction’

Line 315: Replaced ‘was’ with ‘were’

Line 364: Replaced ‘show’ with ‘shows’

Line 513: The dispersal and longevity of neonicotinoids away from point of use is a significant concern, potentially extending the range of pesticide influence.

Line 531: 2015 (added)

Line 588: ‘lack/difficulty’ now explained and written better

Line 647: Re written to ‘… low density and more evenly distributed over the landscape’

Line 679: corrected to ‘trees’

Line 700: Yes, chemical ecology is used in the form of attractants based on ‘herbivore-induced plant volatiles’ (e.g. commercial lures “Predalure”) to bolster natural enemy communities. Ref 134 has the details.

Line 733: Corrected

Line 736: ‘Young leaves’ added to sentence

Line 780: Overwintering monarchs in CA do appear to forage for nectar on occasions, particularly early and late in the overwintering period. I have added water to the sentence.

Line 1138: Removed ‘evolve’ and focused on adapting to changing circumstances.

Reviewer 3 Report

Comments and Suggestions for Authors

James presents an extensive overview on the status of the western North American monarch butterfly population. He begins by documenting the surprising dynamics of this population over  the past ~20 years. He makes the case that elements of these contemporary dynamics are driven primarily by neonicotinoid pesticides, climate change and habitat change. James emphasizes the adaptability of the species, arguing for a prominent role of winter breeding in contributing to the recent resiliency of the population. He goes on to argue that, from a holistic view, other stressors such as human interference and natural enemies are relatively minor contributors to overall population decline. He finally makes a case for continued “human-nature contact” with this species as an important conservation tool. Overall, this review is a timely and important contribution, providing scientists, conservationists, and citizens with key information on all potential causative factors that have led to this decline such that they can take action and determine best practices for conservation. 

James addresses some key points of contention within the monarch community, such as captive-rearing, non-native milkweeds, etc. It is vital to the overall discourse that the arguments in this review be presented. Several minor points of consideration are included to clarify arguments and hopefully strengthen the paper as a whole.

Points of Consideration

-Section 2.1: An explicit indication of how the Thanksgiving and New Years counts are made might be helpful for some readers (i.e. if they are approximated similarly to as is done in Mexico, or if individuals are explicitly counted). 

Lines 94-95: The claim is made that monarchs fluctuated between 200-300,000 from 2001 to 2016. There are six years within this 17 period when counts fell below 200,000. Adjusting the range to 100-300,000 would more accurately reflect the data (15 or 16 of 17 years reach this threshold)

-Figure 1: It will be helpful if this figure included the latest wintering data (through the 2022 counts).

-Section 2.3: The title of this section seems more definitive than is warranted by the information presented. The claim is that neonicotinoid pesticides are the primary driver of population decline post-1997 since habitat loss and climate change are predicted to lead to more gradual declines than are observed. While certainly a reasonable hypothesis, the author acknowledges (e.g. lines 124-127, 215-216) that it is a hypothesis that requires additional study/evidence to more fully support. Therefore, the section title might reflect as much.

-Lines 227-241: It seems that the point of these lines is to support the argument that winter storms were a major contributor to the low overwintering counts measured in 2018 and 2019. The evidence, however, only consists of counts, monarch activity, and storm activity at one location. This point could be more robustly made with data from any additional sites or reference to data on regional storm prevalence. Additional information about the specific example sites would also be helpful. For example, the Santa Cruz Lighthouse Site is the focus here, but we do not have any information about what fraction of the overall population is represented at this site. 

-Lines 309-311: The author states that “the nature of reproductive dormancy has not been explored in western monarchs…” There is “historical” (e.g. Herman 1981,  and others from this author and colleagues) and contemporary literature (e.g. Green 2019) on the Western monarch diapause. The author should correct or clarify this statement based on these papers.

-Lines 315-328: Section two of the paper may benefit from brief analysis/reflection on how different potential population influencers (e.g. temperature, storms, etc.) are integrated. Arguments are presented for causes of specific year’s overwintering counts, but there is little consideration of how these influences may interact. For example, warm temperatures in 2020 are proposed to have tipped monarchs from migratory to non-migratory. However, 2017 and 2019 were nearly just as warm. Would the argument be that increased winter storms during these years resulted in the low numbers? However, if temperatures were warm in these years, monarchs might be predicted to have overwintered in lower frequency. While understanding that this will be speculative, I do think it might be valuable to understand how the author sees how these factors interact.

-Figure 5: I interpret this figure as a main thesis for the review which has the intention to demonstrate how the interaction between all of these stressor, not just one alone, are the drivers of population decline for the western monarchs. However, I think this may be lost on the reader initially given the minimalist nature of the figure, along with a lack of detail (or images/artwork)l to help connect these dots. I suggest this be reworked to more clearly get this thesis across. 

-Lines 569-571: The author may consider adding a summarizing sentence or two on these two references [119,120].

-Lines 738-765: A reader may be interested in the quality of nectar sources along the migratory route (are nectar sources in decline in this area? Might this be a contributor to the decline? A comment may be helpful.

-Lines 826-828: The author states “There is no selection pressure to abandon migration in winter-breeding monarchs, so why would they?” Different lines of evidence suggest that relaxed selection on migration leads to the loss of migration (e.g. Zhan et al., 2014; Pierce et al., 2014; Freedman et al., 2020; Hemstrom et al., 2022) or migratory orientation (e.g. Tenger-Trolander et al. 2019). While the author notes limitations of the latter paper, there does remain indirect evidence from the former ones. Therefore, the author should clarify the original statement or cite additional evidence that may support the claim.

-Lines 1016-1021: The author makes explicit comparisons about the impact of different human interventions. Either references should be cited to support these comparisons, or it should be made explicit that these are speculations (certainly reasonable ones).

-Lines 1050-1052: The author states that “...the Tenger-Trolander studies were flawed by their use of static instead of declining daylengths in attempts to produce migratory adults…” This statement is not completely accurate as Tenger-Trolander and Kronforst (2020) did rear some individuals in this study in a glass greenhouse under natural lighting (i.e. decreasing photoperiod) yet still reported a disruption in orientation.  This statement should be modified.

-Lines 1052-1053: The author states that Tenger-Trolander did not consider that migration can be induced in adult monarchs. This statement should be modified, as Tenger-Trolander and Kronforst (2020) did keep newly eclosed monarchs outdoors for at least 3 days post eclosion. Thus, this study did consider adult migration induction, although perhaps/likely imperfectly. The author presents Wilcox et al. (2020) as a counterargument. However, a reasonable critique to this report was also published (Davis 2020), which might either be acknowledged or challenged. Altogether, it is certainly appropriate to challenge these different studies given their public impact, as the author notes, and these clarifications/corrections should help to make the clearest argument.

-Lines: 1095-1096: The author notes “...that defied logic given the reasons for their life-long incarceration…”. This statement can, and may likely, be construed as offensive and may unnecessarily distract from the author’s well intent to highlight the positive impact of monarch rearing to different communities.

-Lines 1120: Given sensitivity around this term, we ask that the author consider a synonym for “retard”. We acknowledge, however, that the term is absolutely not being used in a derogatory context here and recognize the author’s intended meaning.

Author Response

Response to Reviewer 3

Thank you for your through review and stating that the manuscript is a timely and important contribution to western monarch literature. I also appreciate that you recognize the importance of dealing with contentious issues in the sphere of monarch ecology and conservation and consider it vital that these arguments be presented in the review.

Specifics:

Section 2.1: I believe I have presented an explicit indication of how counts are made in California when I say on Lines 87-92: Citizen monarch counters are trained annually to count monarchs roosting at overwintering sites but in most cases the numbers are good estimates at best because of the density of individuals in clusters. Nevertheless, these ‘counts’ are more accurate than the population estimates of overwintering monarchs in Mexico which are derived from the number of hectares that monarchs occupy [13]. The fact that I say ‘numbers are good estimates at best’ indicates that individuals are not explicitly counted.

Lines 94-95 (or 99-100 on current version). I agree and have made this change.

Figure 1. I have included the 2018-22 data in a new Table.

Section 2.3: I have now added the word ‘potentially’ to the title of this section to make it less definitive.

Lines 227-241: Evidence for winter storms is now enhanced by reference to a Wikipedia report on the 2017 storms. Lighthouse Field is one of the major sites in Santa Cruz as now indicated in the text (line 239).

Lines 315-328: I have changed ‘has not been explored’ to ‘less explored’.

Lines 315-328: I am loathe to speculate too much but have now added a sentence at the end of paragraph 2 in section 207, emphasizing that future populations will be the product of interactions between higher temperatures, increased frequency of storms and adaptation.

Figure 5: Another reviewer suggested this figure was too speculative and unnecessary, so I have removed it.

Lines 569-71: Agreed. I have added some extra text to explain more fully.

Lines 738-828: I have clarified the original statement, emphasizing that there is unlikely to be selection against migration within a single season. Such selection on continuously breeding populations (like commercial insectaries) would only occur after a large number of generations. However, as Freedman et al 2018 showed even continuously breeding populations do retain the ability to convert to migratory physiology/behavior should conditions change.

Lines 1016-1021: I have strengthened the first paragraph of this section making clear, the reasons behind my reasoning. I have toned down all of my assertions using ‘likely’ instead of more definitive wording.

Lines 1050-1052: The butterflies that Tenger-Trolander reared under natural daylengths in their 2020 work, originated from a commercial source. Natural daylengths were not used to rear wild-obtained monarchs in either of their studies (2019, 2020). Thus, my assertion of this flaw in their studies remains correct. I have modified the text to clarify I am talking about wild not commercial-origin monarchs. I have also added a sentence about commercial rearing (and the fact that butterflies from these sources are genetically different) to the paragraph concerning commercially reared monarchs.

Lines 1052-1053: I have modified my statement about adult induction of migration as suggested. I have also added the Davis 2020 ref in regard to his critique of the Wilcox et al. study.

Lines 1095-1096: I have removed this sentence

Line 1120: I have replaced ‘retard’ with ‘inhibit’